# Activation of hedgehog signaling in mesenchymal stem cells induces cartilage and bone tumor formation via Wnt/β-Catenin

Qi Deng[1†], Ping Li[1†], Manju Che[1], Jiajia Liu[1], Soma Biswas[1], Gang Ma[1], Lin He[1], Zhanying Wei[2], Zhenlin Zhang[2], Yingzi Yang[3], Huijuan Liu[1]*, Baojie Li[1,2,4]*

[1]Bio-X Institutes, Key Laboratory for the Genetics of Developmental and Neuropsychiatric Disorders, Shanghai Jiao Tong University, Ministry of Education, Shanghai, China; [2]Metabolic Bone Disease and Genetic Research Unit, Department of Osteoporosis and Bone Diseases, Shanghai Key Clinical Center for Metabolic Disease, Shanghai Jiao Tong University Affiliated Sixth People's Hospital, Shanghai, China; [3]Department of Developmental Biology, Harvard School of Dental Medicine, Boston, United States; [4]State Key Laboratory of Oncogenes and Related Genes, Bio-X-Renji Hospital Research Center, School of Medicine, Renji Hospital, Shanghai Jiao Tong University, Shanghai, China

**Abstract** Indian Hedgehog (IHH) signaling, a key regulator of skeletal development, is highly activated in cartilage and bone tumors. Yet deletion of *Ptch1*, encoding an inhibitor of IHH receptor Smoothened (SMO), in chondrocyte or osteoblasts does not cause tumorigenesis. Here, we show that *Ptch1* deletion in mice Prrx1[+]mesenchymal stem/stromal cells (MSCs) promotes MSC proliferation and osteogenic and chondrogenic differentiation but inhibits adipogenic differentiation. Moreover, *Ptch1* deletion led to development of osteoarthritis-like phenotypes, exostoses, enchondroma, and osteosarcoma in Smo-Gli1/2-dependent manners. The cartilage and bone tumors are originated from Prrx1[+] lineage cells and express low levels of osteoblast and chondrocyte markers, respectively. Mechanistically, *Ptch1* deletion increases the expression of Wnt5a/6 and leads to enhanced β-Catenin activation. Inhibiting Wnt/β-Catenin pathway suppresses development of skeletal anomalies including enchondroma and osteosarcoma. These findings suggest that cartilage/bone tumors arise from their early progenitor cells and identify the Wnt/β-Catenin pathway as a pharmacological target for cartilage/bone neoplasms.
DOI: https://doi.org/10.7554/eLife.50208.001

*For correspondence:
liuhj@sjtu.edu.cn (HL);
libj@sjtu.edu.cn (BL)

[†]These authors contributed equally to this work

## Introduction

The Hedgehog (Hh) signaling pathway controls embryonic pattern formation and organogenesis, adult stem cells homeostasis and tissue maintenance, and is involved in the etiology of various tumors (*Briscoe and Thérond, 2013*). Ligand (Indian, Sonic, or Desert Hedgehog) engagement to receptor Smoothened (Smo) relieves the inhibition of Patched 1 (Ptch1) and upregulates Gli1/2 proteins, which increase the expression of proteins including Myc, Cyclin D, and Bcl2 and promote cell proliferation. Hedgehogs also activate the Rho/Rac pathway and increase the expression of Cyclin B in Smo-independent manners, which are regarded as the non-canonical pathway (*Briscoe and Thérond, 2013*). Human genetic studies have identified germline mutations in *Ptch1* as the cause of Gorlin syndrome, which is characterized by basal cell carcinoma, medulloblastoma, cartilage tumors, and ectopic ossification during adolescence and early adulthood (*Hahn et al., 1996*). Some of the

**eLife digest** Bone and cartilage tumors are among the most common tumors in the skeleton, often affecting the limbs. Bone tumors, also called osteosarcomas, usually occur in growing children and teenagers, and they are often resistant to conventional chemo- and radio-therapies. Surgery is the only treatment option, but this can lead to long-lasting damage that impairs the quality of life of these patients. Thus, there is a need to find new drug targets for these diseases. Unfortunately, no good laboratory-based systems exist that mimic these human cancers, hindering research into these tumors.

One way to create a laboratory-based model for cartilage tumors and osteosarcomas is to reproduce the signaling that is present in the human tumors in a mouse. A signaling pathway called Hedgehog signaling is overactive in human cartilage and bone tumors. The activity of this pathway can be increased by deleting a gene called *Ptch1*; but mice do not form tumors when this gene is deleted in their mature cartilage and bone cells.

Now, Deng, Li et al. report that deleting *Ptch1* in mesenchymal stem cells, early-stage cells that can give rise to cartilage and bone cells, generates a mouse model for osteosarcoma and cartilage tumors. The mice with these *Ptch1* deficient cells developed tumors with overactive Hedgehog signaling in cartilage and bone. Deng, Li et al. also performed biochemical experiments to show that Hedgehog signaling turned on another signaling pathway called Wnt signaling. Treating the mice that had mesenchymal cells lacking *Ptch1* with a drug that inhibits Wnt signaling reduced the growth of cartilage and bone tumors.

These data suggest that deleting *Ptch1* in mouse mesenchymal stem cells can mimic human cartilage tumors and osteosarcomas. More experiments will be needed to explain how the Hedgehog and Wnt signaling pathways interact in these tumors. Finally, further studies will need to investigate if inhibiting Wnt signaling might become a useful therapy for human patients with osteosarcoma in the future.

DOI: https://doi.org/10.7554/eLife.50208.002

patients also develop holoprosencephaly and autism (*Noor et al., 2010*). Inhibitors for Smo or Gli1/2 are developed to treat the related tumors (*Amakye et al., 2013*).

IHH is mainly expressed in prehypertrophic chondrocytes and osteoblasts at puberty stages (*Kindblom et al., 2002*). Genetic studies have shown that IHH signaling regulates proliferation and differentiation of osteoblasts and chondrocytes during skeletal development and repair (*Amano et al., 2015*; *Lanske et al., 1996*; *Maeda et al., 2007*; *Ohba et al., 2008*; *St-Jacques et al., 1999*). IHH regulates chondrocyte proliferation and differentiation mainly via PTHrP (*Lanske et al., 1996*; *Williams et al., 2018*), while IHH regulates osteoblast differentiation by controlling Runx2 expression via the canonical and non-canonical pathways (*Shi et al., 2015*; *Yuan et al., 2016*). Interestingly, Wnt/β-Catenin signaling, a crucial regulator of skeletal development and remodeling, has been shown to mediate the effects of IHH signaling on osteoblast differentiation (*Canalis, 2013*; *Hill et al., 2006*; *Hu et al., 2005*; *Yoshida et al., 2004*), but act upstream of and parallel to IHH signaling in chondrocyte survival and hypertrophy, respectively (*Mak et al., 2006*). In addition, Hh signaling in mature osteoblasts upregulates RANKL expression and enhances osteoclastogenesis and bone resorption (*Mak et al., 2008*). Thus, IHH signaling plays critical roles in skeletal development and remodeling.

Enchondromas and osteosarcomas are among the most common skeleton tumors and they are generally resistant to conventional chemo- and radio-therapies (*Alman, 2015*; *Amakye et al., 2013*; *Kansara et al., 2014*; *Nazeri et al., 2018*). There is an urgent need to identify druggable targets for treatment of these disorders, yet this is hampered by incomplete understanding of pathogenesis of these tumors and a lack of animal models that resemble the human disorders. Cartilage/bone tumors often show activated Hh signaling, resulted either from mutations in *EXT1/2*, *PTH1R*, or *SMO* or from elevated expression of hedgehog ligands or Gli proteins (*Amary et al., 2011*; *Hopyan et al., 2002*; *Pansuriya et al., 2011*; *Tarpey et al., 2013*; *Tiet et al., 2006*). However, activation of Hh signaling, for example by deletion of *Ptch1* alone, in chondrocytes or osteoblasts does not cause tumorigenesis (*Bruce et al., 2010*; *Chan et al., 2014*).

Here, we use *Prrx1-CreERT; Ptch1^{f/f}* mice to study the functions of Hh signaling in mesenchymal stem/stromal cells (MSCs) during adolescence and show that activation of Hh signaling promotes MSC proliferation and osteogenic and chondrogenic differentiation but suppresses MSC adipogenic differentiation and leads to development of osteoarthritis-like phenotypes, enchondroma, and osteosarcoma. *Ptch1* deletion executes these functions via the canonical Hh pathway and activation of Wnt/β-Catenin pathway. This study thus sheds light on the origin of enchondroma and osteosarcoma and identifies Wnt/β-Catenin as a drug target for cartilage/bone tumor treatment.

## Results

### *Ptch1*[+] deletion in Prrx1[+] MSCs in young mice resulted in distorted joints and ectopic bone formation

To investigate the functions of Hh signaling in postnatal bone growth, we ablated *Ptch1* in Prrx1[+] MSCs using inducible *Prrx1-CreERT* mice (*Kawanami et al., 2009*), due to embryonic lethality of *Prrx1-Cre; Ptch1^{f/f}* mice (*Bruce et al., 2010*). Prrx1 has been shown to mark osteoblasts and chondrocytes during development and growth and bone marrow (BM) cells marked by Prrx1 has been shown to have features of MSCs (*Miwa and Era, 2018*). Four doses of tamoxifen (TAM) was intraperitoneally injected into P14 *Prrx1-CreERT; Ptch1^{f/f}* mice, which were euthanized 2 months later, when the peak bone mass was obtained (*Figure 1A*). We found that *Ptch1* deletion led to a decrease in Patched 1 but an increase in Hh target gene *Gli1* in BM-MSCs (*Figure 1B*). *Prrx1-CreERT; Ptch1^{f/f}* mice were significantly smaller and showed decreases in body weight and length compared with age- and gender-matched control mice (*Figure 1C–1E*). Similarly, femur and tibia lengths were decreased by 23.3% and 18.4%, respectively, in *Prrx1-CreERT; Ptch1^{f/f}* mice (*Figure 1F*). X-ray and micro-CT imaging revealed that the mutant mice had deformed knee joints and rough bone surfaces, which are indicative of exostoses (*Figure 1G*). Joint deformation and exostoses were also observed in the phalanges of *Prrx1-CreERT; Ptch1^{f/f}* mice, with the paws being swelled (*Figure 1H*). These results suggest that ablation of *Ptch1* in Prrx1[+] MSCs disrupted bone and cartilage growth and led to joint deformation and ectopic bone growth.

### *Ptch1*[+] deficiency in Prrx1[+] MSCs led to osteoarthritis and enchondroma formation

Histological analyses revealed that *Prrx1-CreERT; Ptch1^{f/f}* mice exhibited signs of osteoarthritis with high OARSI scores at 2.5 months of age (2 months after TAM injection) (*Figure 2A and B*), and increased fibrotic cells lining the damaged cartilage surfaces, which were positive for fibrosis marker FSP1 (*Figure 2C and D*). The articular cartilage showed reduced Col2-expressing cells but increased Col10-expressing cells (*Figure 2E*). We isolated the articular cartilage from the mutant and control mice, carried out quantitative PCR analysis, and detected decreases in *Timp3* (encoding inhibitor of metalloproteinases-3), *Sox9, and Acan* but increases in *Mmp13* (encoding matrix degradation enzyme) and *Adamts5* in the mutant mice (*Figure 2F*). Compared to control littermates, the thickness of calcified cartilage (CC) layer was greater whereas the thickness of hyaline cartilage (HC) layer was lesser in the mutant mice (*Figure 2—figure supplement 1A and B*), accompanied by modest decreases in subchondral bone volume and bone mineral density (BMD) (*Figure 2—figure supplement 1C and D*). These results suggest that *Prrx1-CreERT; Ptch1^{f/f}* mice developed phenotypes resembling early osteoarthritis, consistent with the finding that expression of Smo in Col2[+] chondrocytes leads to development of osteoarthritis (*Lin et al., 2009*).

Histological analyses of bone sections also revealed multiple enchondroma-like lesions (referred to as EC hereafter) at the growth plate, articular cartilage, and bone marrow (likely originated from the growth plate, see late results) in the mutant mice (*Figure 2G*). In addition, chondrocytes in the growth plate of mutant mice was improperly aligned, which also displayed an increase in Ki67 staining in the tumor region and the unaffected region (*Figure 2H and I*). Examination of mice at earlier time points post TAM administration detected EC-like overgrowth at both articular surfaces and growth plates at day 7, which became larger and invaded the trabecular areas over time (*Figure 2—figure supplement 2A*). Expression of Col10 but not Col2 was increased at the growth plate (*Figure 2J and K*). Similar cartilage lesions were also observed in phalanges, humeri, and tibia but

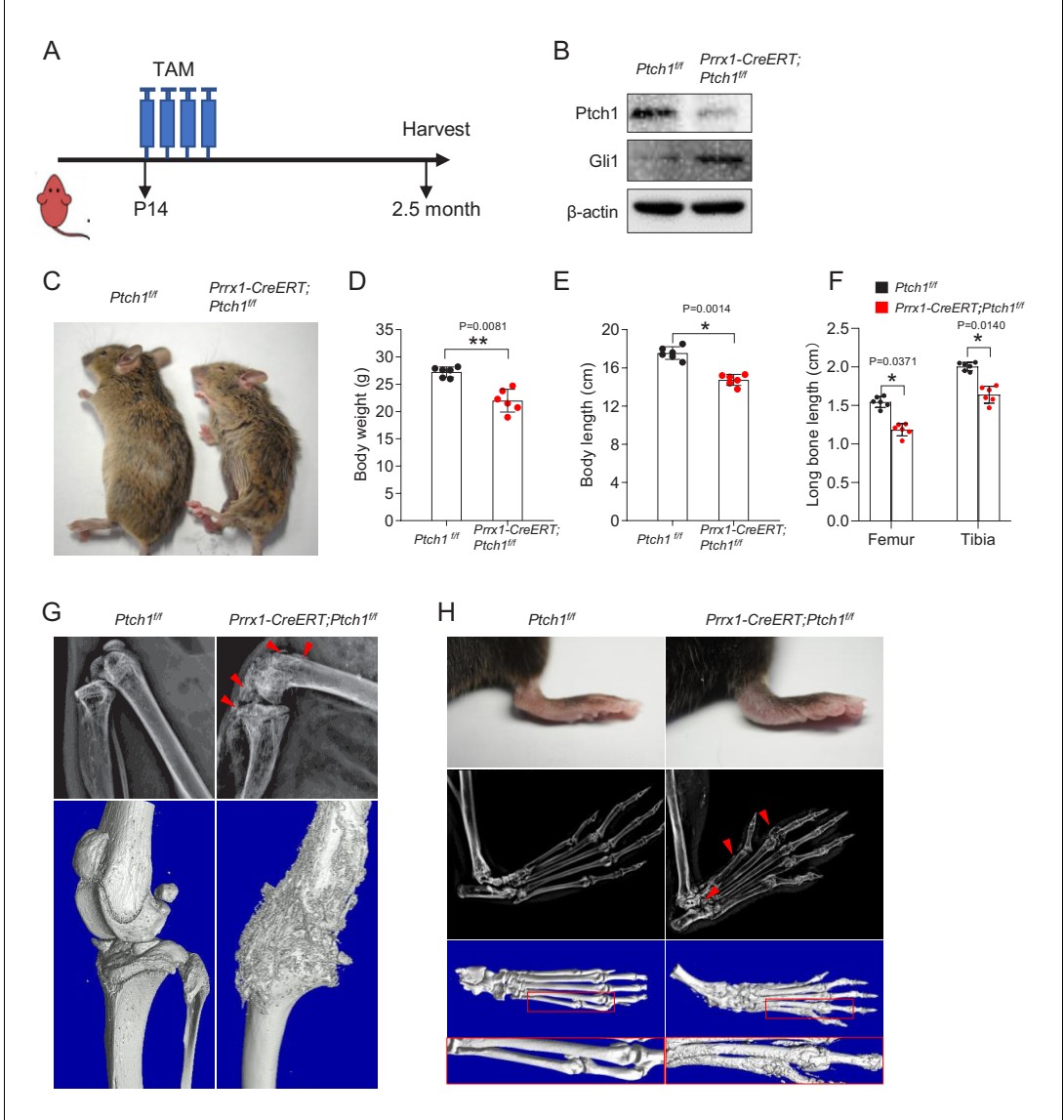

**Figure 1.** Ablation of *Ptch1* in young mice led to joint deformation and exostoses. (**A**) A schematic for the experimental design. (**B**) Western blot results showed that *Ptch1* was largely deleted in BM MSCs and this led to an increase in Gli1 protein. (**C**) *Prrx1-CreERT; Ptch1^{f/f}* mice appeared smaller than control littermates 2 months after TAM injection. (**D**) *Prrx1-CreERT; Ptch1^{f/f}* mice showed a decrease in body weight compared with control littermates, **p<0.01, n = 6. (**E**) *Prrx1-CreERT; Ptch1^{f/f}* mice showed a decrease in body length compared with control littermates, *p<0.05, n = 6. (**F**) *Prrx1-CreERT; Ptch1^{f/f}* mice showed a decrease in femur length compared with control littermates, *p<0.05, n = 6. (**G**) Radiographic and micro-CT images of hindlimb showed joint deformation and exostoses in *Prrx1-CreERT; Ptch1^{f/f}* mice. (**H**) Radiographic and micro-CT images of the paws showed joint deformation and exostoses in *Prrx1-CreERT; Ptch1^{f/f}* mice.

DOI: https://doi.org/10.7554/eLife.50208.003

not in the vertebrae of mutant mice (*Figure 2—figure supplement 2B–2F*). Overall, *Ptch1* deletion in Prrx1+ MSCs promoted chondrocyte proliferation and enchondroma formation.

### *Ptch1* deficiency led to osteosarcoma formation at periosteal surfaces

*Prrx1-CreERT;Ptch1^{f/f}* mice also developed tumors at the periosteal surfaces that have features of osteosarcoma: expansive osteoid lesions with mushroom-shaped appearance that were only located at cortical bones of the limbs (*Figure 3A–3D*), which later transgressed the cortex (*Figure 3—figure supplement 1A*). Note that exostoses, which were smaller and numerous, were observed in long bones and phalanges of *Prrx1-CreERT; Ptch1^{f/f}* mice (*Figure 3—figure supplement 1B*).

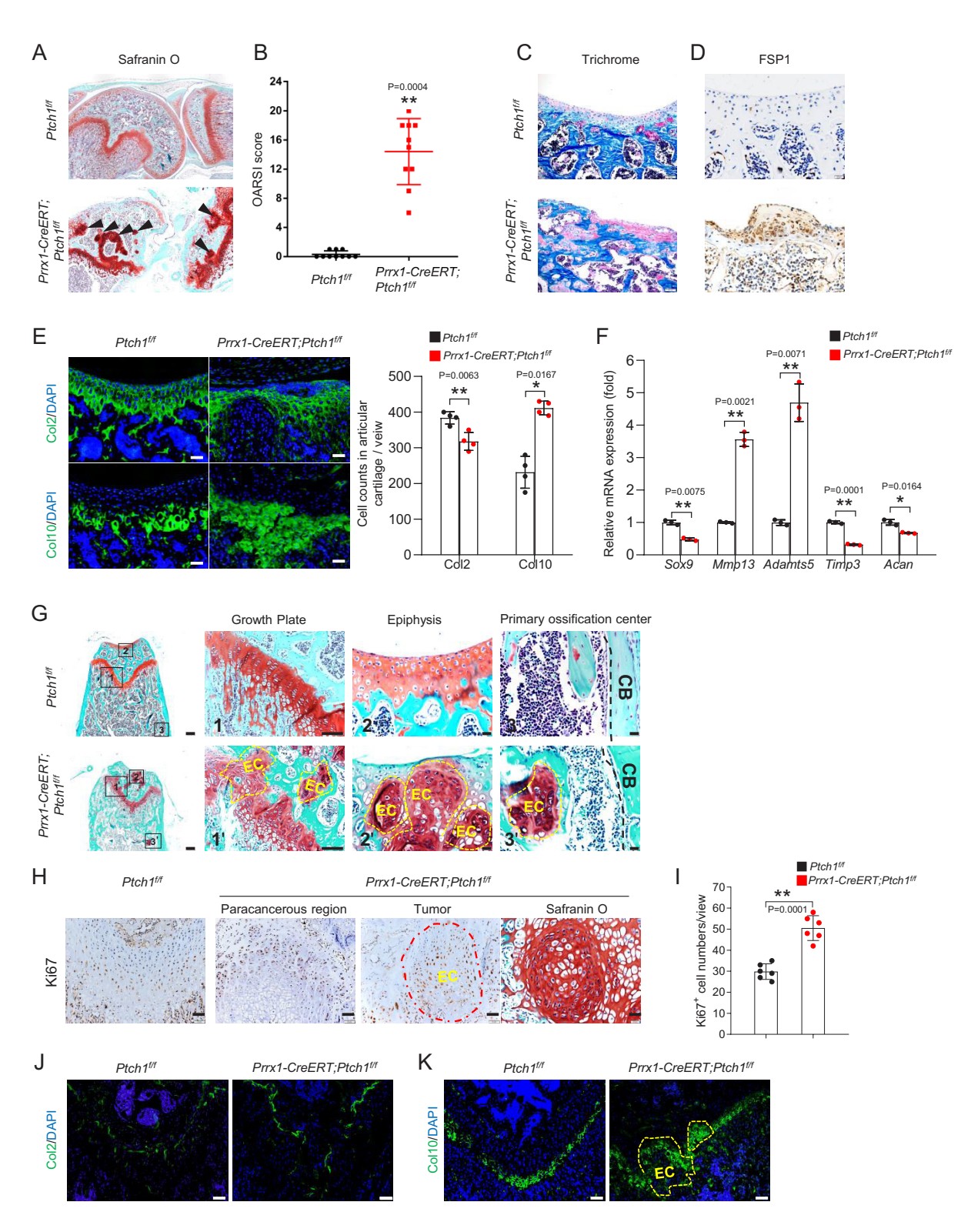

**Figure 2.** Ablation of *Ptch1* led to increased chondrocyte proliferation and development of osteoarthritis and enchondroma. (A) Safranin O staining of knee joints of *Prrx1-CreERT; Ptch1^f/f* and control mice. (B) Increased OARSI scores of *Prrx1-CreERT; Ptch1^f/f* mice compared to control littermates. **p<0.01 compared to the *Ptch1^f/f* group, n = 9 per group. (C) Villanueva-Goldner's trichrome staining of articular cartilage of *Prrx1-CreERT; Ptch1^f/f* and control mice. (D) Immunostaining for FSP1 of articular cartilage of *Prrx1-CreERT; Ptch1^f/f* and control mice. (E) Immunostaining of Col2 (top) and

*Figure 2 continued on next page*

Figure 2 continued

Col10 (bottom) on the articular cartilage of femur bones in *Prrx1-CreERT; Ptch1^{f/f}* and control mice. Right panel: quantitation data, *p<0.05, **p<0.01, n = 4. (F) Quantitative PCR analysis of articular cartilage samples of *Prrx1-CreERT; Ptch1^{f/f}* and control mice revealed alteration of gene expression that indicated osteoarthritis phenotypes. All samples were normalized to GAPDH and then to the control samples, *p<0.05, **p<0.01, n = 3. (G) Safranin O staining showed multiple enchondroma-like lesions (EC) at growth plate, articular cartilage, and bone marrow in *Prrx1-CreERT; Ptch1^{f/f}* mice. Higher magnification views of boxed areas were shown on the right. CB, cortical bone. (H) Ki67 immunohistochemistry at the growth plate of femurs of *Prrx1-CreERT; Ptch1^{f/f}* and control mice. Right panel: Safranin O staining of section corresponding to tumor region. (I) Quantitation of Ki67$^+$ cells at the growth plate of femur bones (non-tumor regions) of *Prrx1-CreERT; Ptch1^{f/f}* and control mice, **p<0.01, n = 6. (J) Immunostaining of Col2 at the growth plate of femur in *Prrx1-CreERT; Ptch1^{f/f}* and control mice. (K) Immunostaining of Col10 at the growth plate of femur in *Prrx1-CreERT; Ptch1^{f/f}* and control mice.

DOI: https://doi.org/10.7554/eLife.50208.004

The following figure supplements are available for figure 2:

**Figure supplement 1.** OA-like changes of subchondral bone architecture in *Prrx1-CreERT; Ptch1^{f/f}* mice.
DOI: https://doi.org/10.7554/eLife.50208.005
**Figure supplement 2.** Histological analysis of the skeleton of *Prrx1-CreERT; Ptch1^{f/f}* mice.
DOI: https://doi.org/10.7554/eLife.50208.006

Osteosarcoma-like lesions (referred to as OS hereafter) also showed increased angiogenesis, with the blood vessels mainly located outside of the tumors (*Figure 3—figure supplement 1C*). All (n = 9) *Prrx1-CreERT; Ptch1^{f/f}* mice developed OS and they lived less than 7 months after TAM injection. Two of the mutant mice showed tumors in the lung (*Figure 3—figure supplement 1D*), indicative of metastasis.

Quantitative PCR and western blot analyses confirmed that Hh signaling was activated in the OS tumors, manifested by increases in the expression of Gli1 protein and other IHH target genes compared to cultured periosteal cells (*Figure 3E and F*). We also isolated primary cells from osteosarcomas and cultured them for further analysis. Immunostaining results showed that there were more Ki67-poisitve cells in osteosarcoma cell cultures than normal periosteal cell cultures (*Figure 3G*). A significant increase in cell proliferation rate was also observed in osteosarcoma cells (*Figure 3H*). Wound healing assays revealed that osteosarcoma cells showed a significant increase in cell migration rate compared with periosteal cells (*Figure 3I*).

However, micro-CT analysis revealed no significant change in bone mass and the number or thickness of trabecular bones in *Prrx1-CreERT; Ptch1^{f/f}* mice at 2.5 months of age (*Figure 3—figure supplement 2*). This could be due to the tight coupling between osteoblastogenesis and osteoclastogenesis caused by Hh activation (*Mak et al., 2008*).

## *Ptch1* deletion in Gli1$^+$ chondrocytes/osteoprogenitors did not lead to tumor formation

The above results are in contrast to a previous study showing that *Col2α1-Cre*-mediated *Ptch1* deletion only led to delayed chondrocyte hypertrophy (*Mak et al., 2006*). We also deleted *Ptch1* using *Gli1-CreERT* mouse, which was reported to label osteoprogenitors (underneath the growth plate) and chondrocytes (*Shi et al., 2017*). We found that the mutant line did not develop tumors at all (*Figure 3—figure supplement 3*), yet, they showed deceases in trabecular bones (*Figure 3—figure supplement 3B*). The lack of tumor formation in *Gli1-CreERT; Ptch1^{f/f}* mice suggests that activation of Hh signaling in MSCs but not in chondrocytes or osteoprogenitors promotes tumorigenesis. Note that deletion of one allele of *Gli1*, like in *Gli1-CreERT* mice, does not affect *Ptch1* deficiency-induced tumorigenesis (*Kimura et al., 2005*).

## *Ptch1* deletion produced skeletal phenotypes via the canonical Smo-Gli pathway

To test the contribution of canonical Hh signaling to *Ptch1* deficiency-induced skeletal defects and tumorigenesis, we treated *Prrx1-CreERT; Ptch1^{f/f}* mice with Smo inhibitor cyclopamine or Gli1/2 inhibitor GANT61 for 2 months after TAM administration. X-ray and histological examination revealed that cyclopamine or GANT61 alleviated joint deformation and enchondroma formation, and restored the structures of articular cartilage and growth plate (*Figure 4A–4C*). Cyclopamine or GANT61 also diminished the development of osteosarcoma in *Prrx1-CreERT; Ptch1^{f/f}* mice

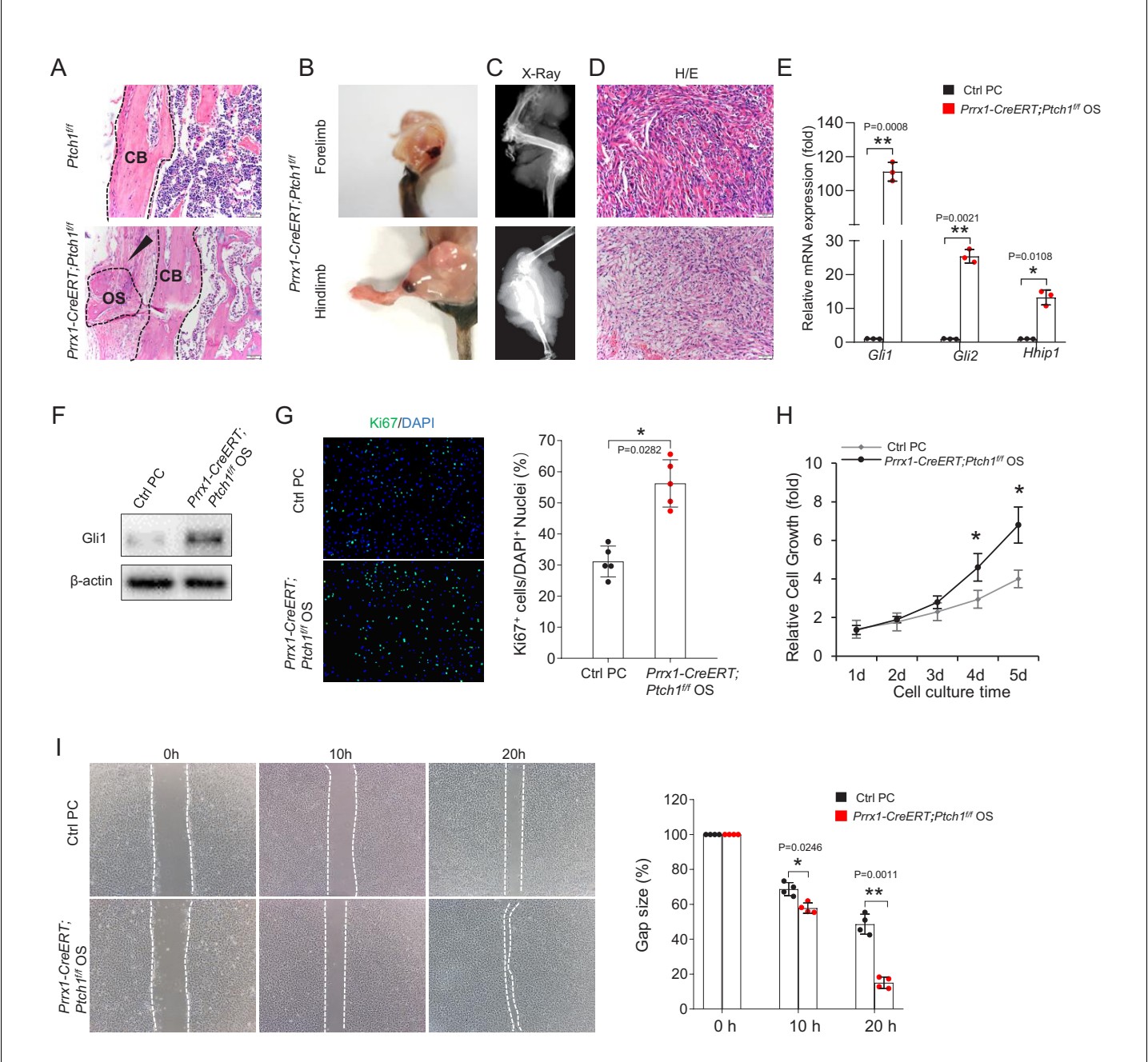

**Figure 3.** *Prrx1-CreERT; Ptch1^f/f* mice developed osteosarcoma. (**A**) H/E staining showed that osteoid lesions (arrows) were present in femur of *Prrx1-CreERT; Ptch1^f/f* mice. OS, osteosarcoma; CB, cortical bone. (**B**) Representative presentation of osteosarcoma (OS) in forelimb (top) and hindlimb (bottom) in *Prrx1-CreERT; Ptch1^f/f* mice 5 months after TAM injection. (**C**) Radiographic images of bone tumors in *Prrx1-CreERT; Ptch1^f/f* mice 5 months after TAM injection. (**D**) Representative histological section of osteosarcoma in *Prrx1-CreERT; Ptch1^f/f* mice 5 months after TAM injection. (**E**) Quantitative PCR analysis of Hh target genes of tumor tissues isolated from *Prrx1-CreERT; Ptch1^f/f* mice. All samples were normalized to GAPDH and then to control periosteal cells (PC), *p<0.05, **p<0.01, n = 3. (**F**) Western blot showed an increase in Gli1 in bone tumor tissues isolated from *Prrx1-CreERT; Ptch1^f/f* mice compared to normal periosteal cells. (**G**) Cell proliferation assays for primary osteosarcoma cells and control periosteal cells by Ki67 staining, Right panel: quantitation data. *p<0.05, n = 5. (**H**) Cell proliferation analysis of primary osteosarcoma cells and control periosteal cells using CKK8 assays, *p<0.05, n = 3. (**I**) Wound healing assay of primary osteosarcoma cells. Dotted lines indicated the cell fronts, Right panel: quantitation data. *p<0.05, **p<0.01, n = 4.

DOI: https://doi.org/10.7554/eLife.50208.007

The following figure supplements are available for figure 3:

**Figure supplement 1.** Enhanced angiogenesis and possible metastasis in *Prrx1-CreERT; Ptch1^f/f* mice.

*Figure 3 continued*

DOI: https://doi.org/10.7554/eLife.50208.008

**Figure supplement 2.** Activation of Hh signaling in MSCs did not alter bone mass.

DOI: https://doi.org/10.7554/eLife.50208.009

**Figure supplement 3.** Deletion of *Ptch1* in Gli1+ cells did not lead to development of enchondroma or osteosarcoma.

DOI: https://doi.org/10.7554/eLife.50208.010

(*Figure 4D and E*). Overall, these findings indicate that Hh signaling regulates skeletal growth and

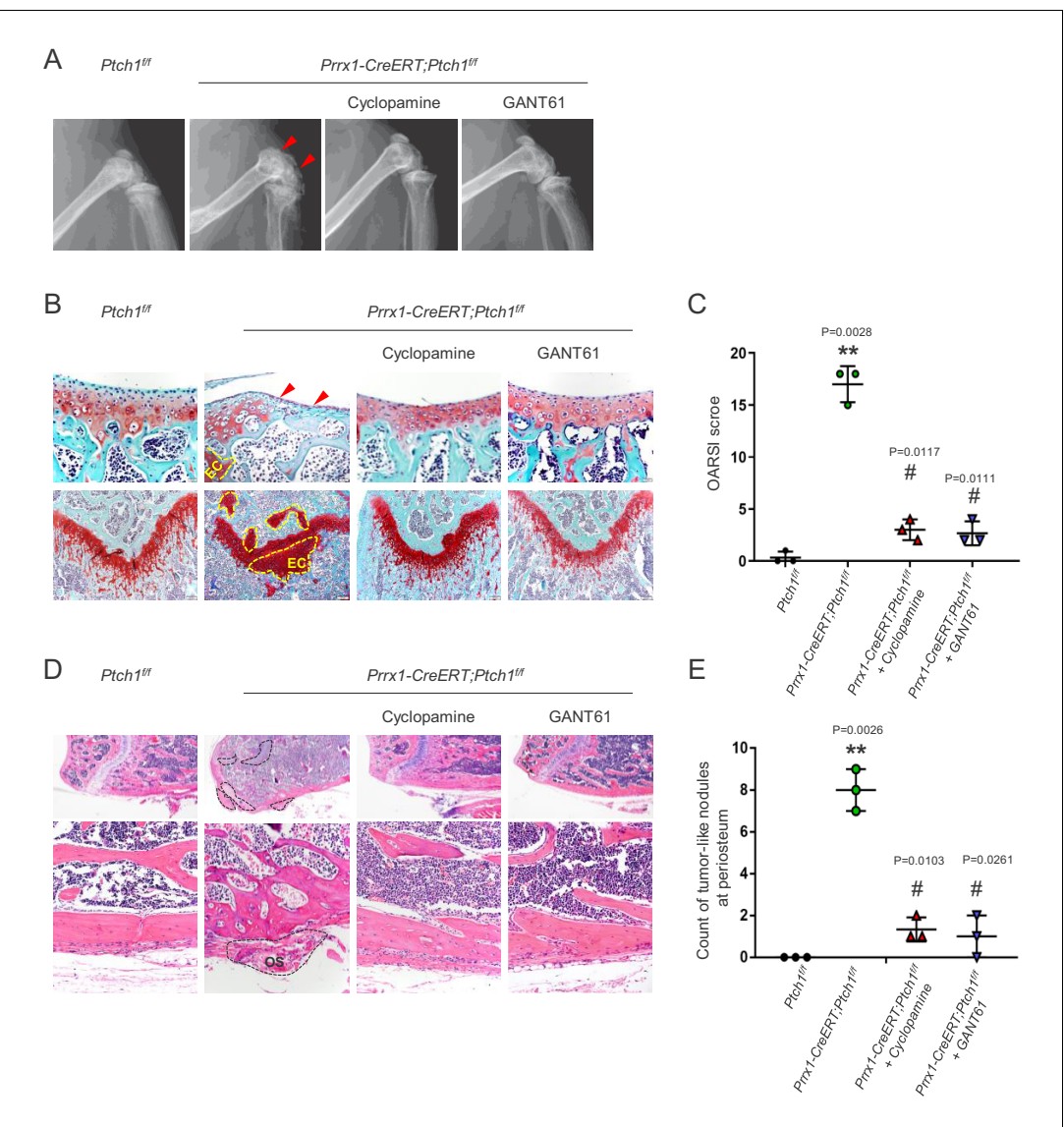

**Figure 4.** *Ptch1* deficiency induced cartilage/bone tumor formation via Smo-Gli1/2. (**A**) Radiographic images of *Prrx1-CreERT; Ptch1*f/f and control mice treated with cyclopamine or GANT61. (**B**) Safranin O staining showed that Cyclopamine or GANT61 rescued the disrupted articular cartilage and formation of enchondroma in *Prrx1-CreERT; Ptch1*f/f mice. (**C**) OARSI scores of control and *Prrx1-CreERT; Ptch1*f/f mice treated with either vehicle, Cyclopamine, or GANT61. \*\*p<0.01 compared to the *Ptch1*f/f group; # p<0.05 compared to the vehicle-treated *Prrx1-CreERT; Ptch1*f/f group, n = 3 per group. (**D**) H/E staining showed that Cyclopamine or GANT61 rescued the osteosarcoma-like lesions in *Prrx1-CreERT; Ptch1*f/f mice. (**E**) Counts of osteogenic tumor-like nodules in control and *Prrx1-CreERT; Ptch1*f/f mice treated with either vehicle, Cyclopamine, or GANT61. \*\*p<0.01 compared to the *Ptch1*f/f group; # p<0.05 compared to the vehicle-treated *Prrx1-CreERT; Ptch1*f/f group, n = 3 per group.

DOI: https://doi.org/10.7554/eLife.50208.011

promotes tumorigenesis via the Smo-Gli1/2 pathway.

## The origins of *Ptch1* deficiency-induced enchondroma and osteosarcoma

The above results indicate that deletion of *Ptch1* alone in Prrx1[+] MSCs resulted in development of both enchondroma and osteosarcoma in the same mouse. To trace the origin of these tumors, we generated *Prrx1-CreERT;Ai14* and *Prrx1-CreERT; Ptch1^{f/f}; Ai14* mice. Lineage tracing revealed that immediately after TAM injection, a few Ai14[+] cells were detected in the articular cartilage, growth plate, periosteum, and large numbers of Ai14[+] cells in the trabecular bones, whereas *Prrx1-CreERT; Ai14* mice without TAM administration showed no Ai14[+] cells (*Figure 5A* and *Figure 5—figure supplement 1A*). Over time, Ai14[+] cells were expanded and replenished articular cartilage, growth plate, BM, and periosteum (*Figure 5A*), suggesting that Prrx1 may label stem/progenitor cells at these locations. *Ptch1* deletion led to further expansion of Prrx1 lineage cells (*Figure 5A*). Furthermore, Ai14[+] enchondroma (at articular cartilage and growth plate) and OS-like lesions (at periosteal bone surfaces) were detected in *Prrx1-CreERT; Ptch1^{f/f}; Ai14* mice at days 14 and 30 post TAM administration, respectively (*Figure 5A*), suggesting that *Ptch1* deletion quickly leads to overproliferation of MSCs or progenitors. More enchondromas and osteosarcomas were formed 2 months post TAM injection (*Figure 5B*). The growth plate did not show much differences in the thickness of different zones (*Figure 5—figure supplement 1B*). These results suggest that the tumors are originated from Prrx1 lineage cells located at articular cartilage, growth plate, and periosteal bones but not at trabecular bones. On the other hand, Prrx1 marked limited numbers of osteoblasts and no chondrocytes in the vertebrae (*Figure 5—figure supplement 1C*), consistent with our observation that *Prrx1-CreERT; Ptch1^{f/f}* mice did not develop tumors in vertebrae (*Figure 2—figure supplement 2E*).

## Evidence that enchondroma and osteosarcoma arise from chondrocyte and osteoblast progenitors, respectively

Previous studies have shown that Prrx1-marked BM cells have osteoblast, chondrocyte, and adipocyte differentiation potentials (*Miwa and Era, 2018*). We found Prrx1-marked BM-MSCs accounted for more than 50% of the adherent MSCs (*Figure 5—figure supplement 2A*). Moreover, Prrx1[+] cells isolated from periosteal surfaces of *Prrx1-CreERT;Ai14* mice (right after four daily doses of TAM) could form colony-forming units and had osteoblast, chondrocyte, and adipocyte differentiation potentials (*Figure 5—figure supplement 2B and C*). Prrx1[+] cells could also differentiate into chondrocytes and osteoblasts during bone fracture repair in vivo (data not shown). Overall, these results suggest that Prrx1[+] cells in bone marrow and periosteal bones are multipotent. Note that our tracing data indicate that enchondromas observed in BM cavities are originated from multipotent cells located in the growth plate but not BM MSCs.

Immunostaining revealed that Ai14[+] enchondroma expressed low levels of Col2 and Col10, but not Col1 (*Figure 5C*), whereas Ai14[+] osteosarcoma cells, which were located at periosteal surfaces, expressed low levels of Col1 but not Col2 or Col10 (*Figure 5C* and *Figure 5—figure supplement 1D*), suggesting that these tumor cells were in a low-degree of differentiation state and that enchondroma and osteosarcoma arise from early chondrocyte and osteoblast progenitors, respectively. It is predicted that tumors derived from multipotent MSCs would contain both osteoblasts and chondrocytes, yet no such tumor was detected in *Prrx1-CreERT; Ptch1^{f/f}* mice.

## Hh signaling promotes Prrx1[+] MSC proliferation and differentially regulates its tri-lineage differentiation

We found that BM-MSCs from 2.5 month-old *Prrx1-CreERT; Ptch1^{f/f}* mice showed an increase in colony forming units (*Figure 6A*), indicating that *Ptch1* ablation led to an expansion of the BM-MSC pool in vivo and/or increased cell proliferation. BM-MSCs isolated from *Prrx1-CreERT; Ptch1^{f/f}; Ai14* mice indeed showed increased proliferation rates, manifested by an increase in Ki67-positive cells (*Figure 6B*), enhanced osteogenic and chondrogenic differentiation, manifested by increased histological staining (*Figure 6C*). We found that overexpression of *Ptch1* in BM-MSCs suppressed osteoblast and chondrocyte differentiation (*Figure 6—figure supplement 1A–1C*), demonstrating the negative roles for *Ptch1* in osteoblast and chondrocyte differentiation. *Ptch1*-deficient BM-MSCs

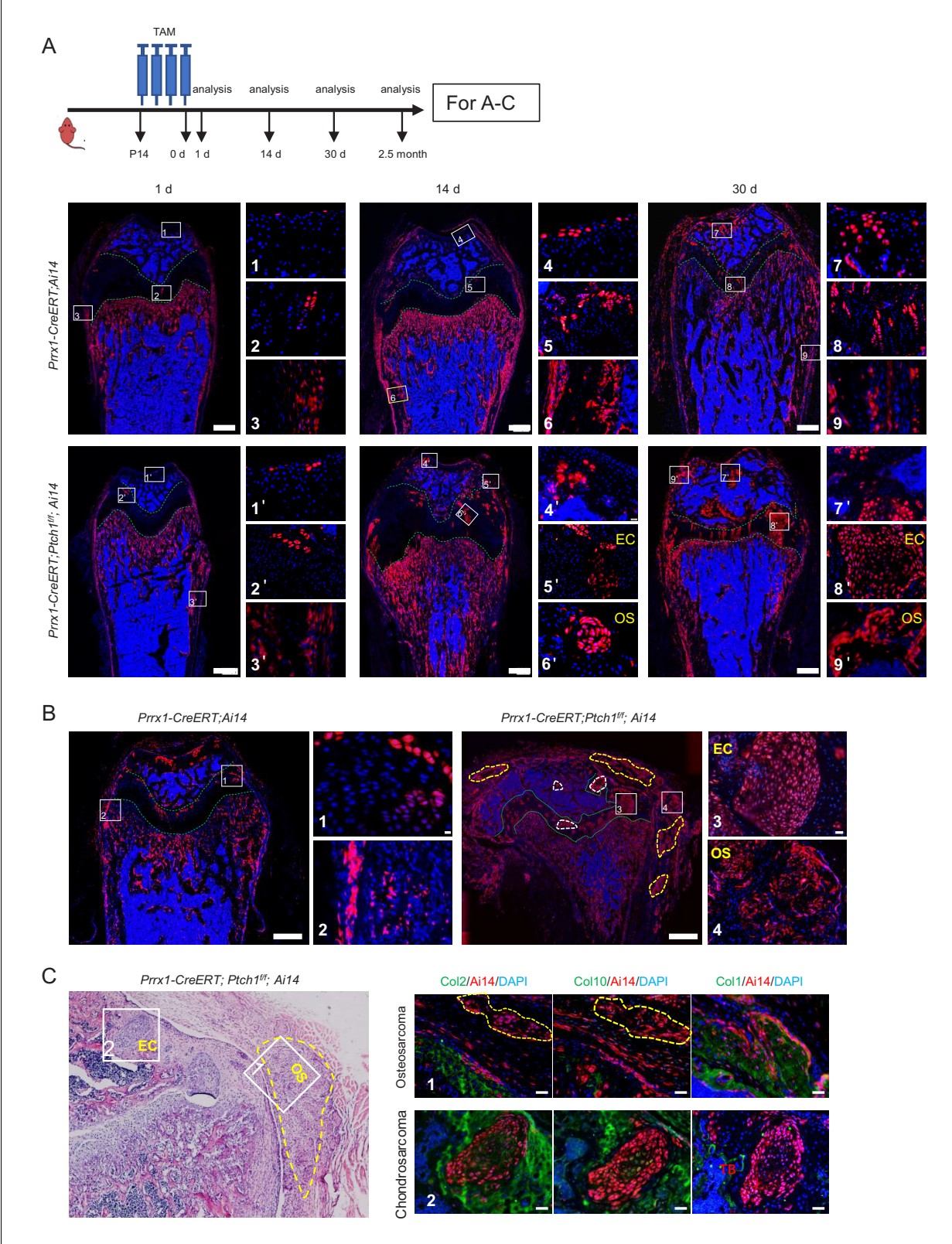

**Figure 5.** *Ptch1* deficiency-induced cartilage/bone tumors were originated from Prrx1 lineage cells. (**A**) Tracing results of 1, 14, and 30 days after 4 doses of TAM injection in *Prrx1-CreERT;Ptch1^{f/f}; Ai14* and *Prrx1-CreERT; Ai14* mice. EC, enchondroma; OS, osteosarcoma. Upper panel: a schematic for the experimental design. (**B**) Lineage tracing of *Prrx1-CreERT; Ptch1^{f/f};Ai14* and *Prrx1-CreERT;Ai14* mice 2.5 months after TAM injection. Images showed

*Figure 5 continued on next page*

*Figure 5 continued*

that cells in the tumor region are Prrx1⁺ (red). White circles, enchondroma; yellow circles, osteosarcoma. EC, enchondroma; OS, osteosarcoma. (C) Immunostaining of chondrocyte and osteoblast markers on bone tumor sections. Left panel: H/E staining.

DOI: https://doi.org/10.7554/eLife.50208.012

The following figure supplements are available for figure 5:

**Figure supplement 1.** Marking of growth plate and vertebrae by Prrx1 in *Prrx1-CreERT;Ptch1^(f/f)* mice.

DOI: https://doi.org/10.7554/eLife.50208.013

**Figure supplement 2.** Prrx1⁺periosteal cells showed features of MSC.

DOI: https://doi.org/10.7554/eLife.50208.014

also showed suppressed adipogenic differentiation, manifested by reduced levels of Oil Red O staining (*Figure 6C*), which is supported by the observation that *Prrx1-CreERT; Ptch1^(f/f)* mice showed a decrease in Perilipin⁺ adipocytes in the bone marrow (*Figure 6—figure supplement 2*). Analyses of expression of osteoblast, chondrocyte, and adipocyte markers confirmed these histological staining results (*Figure 6D*). We validated the effects of *Ptch1* ablation on MSC differentiation by transplantation assays. Carrier particles of hydroxyapatite tricalcium phosphate (HA/TCP) were mixed with MSCs and implanted subcutaneously under the dorsal skin of nude mice. After 8 weeks, the implants were harvested and analyzed. It was found that mutant MSCs formed more bone and fewer adipocytes in vivo than control MSCs (*Figure 6E*). We also used a cell pellet culture model to confirm the effects of *Ptch1* ablation on chondrogenic differentiation. MSCs isolated from mutant and control mice were pelleted by centrifugation and maintained in chondrogenic culture medium for 3 weeks. We found that the mutant MSCs showed increases in chondrocyte pellet size, which may be attributable to increased hypertrophic growth and increased proliferation, which were manifested by H/E, toluidine blue, and Ki67 staining (*Figure 6F and G*). Taken together, these results suggest that Hh signaling plays critical roles in MSC proliferation and differentiation.

## *Ptch1* deficiency led to elevated expression of *Wnt5a* and *Wnt6*

We next searched for possible downstream mediators of *Ptch1* deficiency-induced tumorigenesis and other skeletal phenotypes. We compared activation of important signaling molecules that regulate chondrocyte and osteoblast proliferation and differentiation, in BM MSCs isolated from *Prrx1-CreERT; Ptch1^(f/f)* and control mice by western blot. We found that *Ptch1* deficiency increased the activation of β-Catenin and Akt1 but suppressed the activation of Smad1/5/8 without affecting activation of Smad2/3 or Erk1/2 (*Figure 7A*). Immunostaining of bone sections also confirmed the changes in activation of these pathways (*Figure 7B*). *Ptch1* deficiency-induced increase in β-Catenin occurred in both the nucleus and cytoplasm ( Figure 7-figure supplement 1A). Furthermore, Hh signaling can activate β-Catenin in wildtype MSCs (*Figure 7—figure supplement 1B and C*). Previous studies have reported that Hh signaling interacts with BMP and WNT pathways to regulate cell fate determination (*Zhao et al., 2006*). However, suppressed Smad1/5/8 activation could not explain increased MSC differentiation into osteoblasts and chondrocytes.

Since the Wnt/β-Catenin pathway promotes proliferation and differentiation of both osteoblasts and chondrocytes, we determined the expression of 19 Wnt molecules in *Ptch1^(-/-)*BM MSCs. Quantitative PCR analysis showed that *Wnt5a* and *Wnt6* were expressed at significantly higher levels in *Ptch1^(-/-)*BM MSCs than control cells (*Figure 7C*). To test whether *Wnt5a* and *Wnt6* are target genes of Hh signaling, we performed chromatin-immunoprecipitation (ChIP) assays for Hh downstream transcription factor Gli1. We found that both *Wnt5a* and *Wnt6* contained two binding sites for Gli1 and moreover, binding of Gli1 to these sites were increased in *Ptch1* deficient cells (*Figure 7D–7F*). These results suggest that Hh signaling may activate transcription of *Wnt5a* and *Wnt6* in MSCs via Gli-1.

## Human cartilage/bone tumor samples showed activation of both Hh and Wnt pathways

The above mouse studies established a link between Hh signaling and Wnt/β–Catenin signaling in skeletal growth and enchondroma/osteosarcoma development. Wnt/β–Catenin activation is a common event in various human tumors, especially in colorectal cancer (*Anastas and Moon, 2013*). We

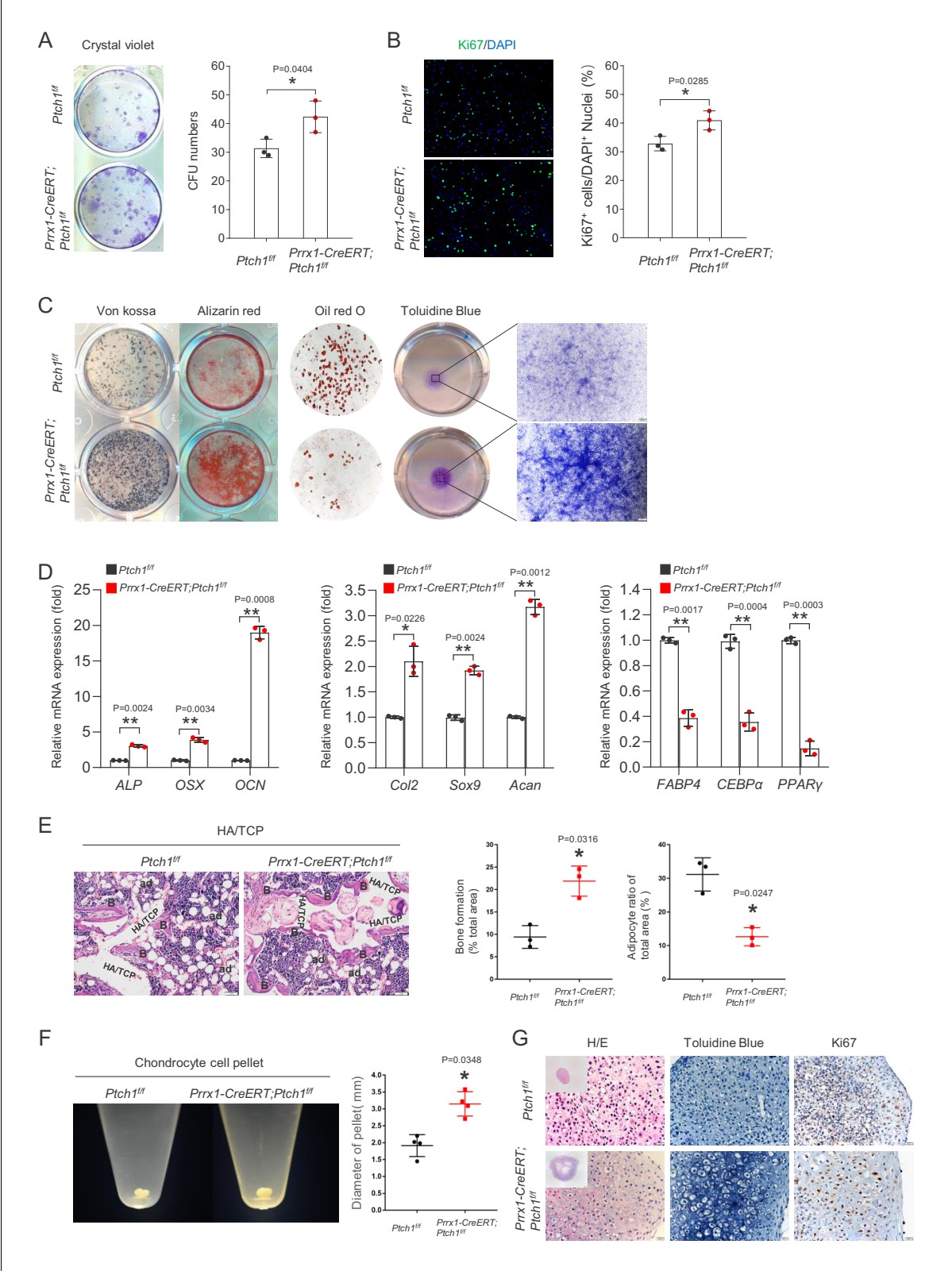

**Figure 6.** Critical roles for Hh signaling in MSC proliferation and differentiation. (**A**) Colony forming efficiency assay of BM MSCs isolated from *Prrx1-CreERT; Ptch1^{f/f}* and control mice. These plates were stained with crystal violet. Right panels, quantitation data, *p<0.05, n = 3. (**B**) Cell proliferation of MSCs isolated from *Prrx1-CreERT; Ptch1^{f/f}* and control mice by Ki67 staining. Right panel: quantitation data. *p<0.05, n = 3. (**C**) *Prrx1-CreERT; Ptch1^{f/f}* MSCs showed an alteration in tri-lineage differentiation activities. Osteoblast differentiation was judged by Von Kossa and Alizarin Red staining, *Figure 6 continued on next page*

*Figure 6 continued*

chondrocyte differentiation was judged by Toluidine Blue staining, and adipocyte differentiation was judged by Oil Red O staining. (D) Quantitative PCR analysis of lineage-specific markers of *Ptch1*-/- and control MSCs cultured with osteoblast medium, chondrocyte medium or adipocyte medium. All samples were normalized to GAPDH and then to the controls, *p<0.05, **p<0.01, n = 3. (E) H/E staining of histological sections from implanted *Ptch1*-/- and control MSC-scaffolds. Right panel, quantitative analysis of amount of bone and adipocytes that formed on the HA/TCP particles using Image-Pro Plus software based on H/E staining. HA/TCP, hydroxyapatite/tricalcium phosphate; B, bone; ad, adipocytes. Right panel: quantitation data. *p<0.05, n = 3. (F) Chondrocyte cells pellet images of differentiated *Ptch1*-/- and control BM MSCs. Right panel, quantitative data of pellet sizes. *p<0.05, n = 4. (G) Chondrocyte cell pellet assays of differentiation of *Ptch1*-/- and control BM MSCs. The same numbers of mutant and control MSCs were induced to differentiate into chondrocytes and the section were stained with H/E (left), toluidine blue (middle), or Ki67 (right).

DOI: https://doi.org/10.7554/eLife.50208.015

The following figure supplements are available for figure 6:

**Figure supplement 1.** Overexpression of Ptch1 inhibited BM-MSC osteoblast and chondrocyte differentiation.

DOI: https://doi.org/10.7554/eLife.50208.016

**Figure supplement 2.** Adipogenesis was decreased in femurs of *Prrx1-CreERT; Ptch1*f/f mice.

DOI: https://doi.org/10.7554/eLife.50208.017

then tested whether Wnt/β–Catenin pathway was activated in human cartilage/bone tumors. Immunohistochemical staining of 24 human cartilage/bone tumors revealed that β–Catenin levels were increased in most of the tumor samples compared with normal tissues (*Figure 7G and H* and *Figure 7—figure supplement 2*), which was correlated with increased levels of Gli1 (*Figure 7H*). These results suggest that the link between Hh and Wnt/β–Catenin pathways also exists in human tumor samples.

## Inhibition of Wnt/β-Catenin suppressed MSC proliferation and differentiation

We then tested whether the Wnt/β-Catenin pathway mediated the effects of *Ptch1* deficiency on MSC proliferation and chondrogenic/osteogenic differentiation. To this end, we used IWP2 to inhibit Wnt/β-Catenin activation in MSC cultures. IWP2 is a small molecule compound that specifically inhibits Wnt signaling and has been used in many studies (*Chen et al., 2009*; *Jeong et al., 2017*). We found that accelerated MSC proliferation caused by *Ptch1* ablation was markedly suppressed by IWP2 but not by BMP2 (*Figure 8A*). In addition, IWP2 blunted accelerated osteogenic and chondrogenic differentiation of *Ptch1*-/-MSCs while BMP2 enhanced the differentiation (*Figure 8B*). In addition, FH535, another Wnt/β-Catenin inhibitor, also suppressed accelerated osteogenic and chondrogenic differentiation of *Ptch1*-/-MSCs (*Figure 8—figure supplement 1*). Overall, these results indicate that Wnt/β-Catenin mediates accelerated MSC proliferation and differentiation caused by *Ptch1* deletion.

## Inhibition of Wnt/β-Catenin suppressed tumor formation in *Prrx1-CreERT; Ptch1*f/f mice

Encouraged by the in vitro results on the effects of IWP2 on MSC proliferation and differentiation, we tested whether IWP2 could rescue the skeletal phonotypes observed in *Prrx1-CreERT; Ptch1*f/f mice. We treated *Prrx1-CreERT; Ptch1*f/f mice with IWP2 for 2 months right after TAM administration. IWP2 caused a decrease in the β–Catenin levels on bone sections without affecting the levels of Gli1 (*Figure 8C*). Importantly, IWP2 treatment rescued joint deformation and growth plate defects and diminished enchondroma and osteosarcoma formation of *Prrx1-CreERT; Ptch1*f/f mice (*Figure 8D–8H* and *Figure 8—figure supplement 2*). Overall, these results suggest that enhanced Wnt/β–Catenin signaling is responsible for cartilage/bone growth defects and tumor development caused by *Ptch1* deficiency in MSCs.

## Discussion

This study for the first time shows that deletion of *Ptch1* alone in Prrx1+ MSCs results in development of enchondromas and osteosarcomas in the same mice. This is in stark contrast to deletion of *Ptch1* in osteoblasts or chondrocytes and suggests that stemness or differentiation status of the cell plays a critical role in cartilage/bone tumorigenesis. We further show that *Ptch1*-/-enchondromas and

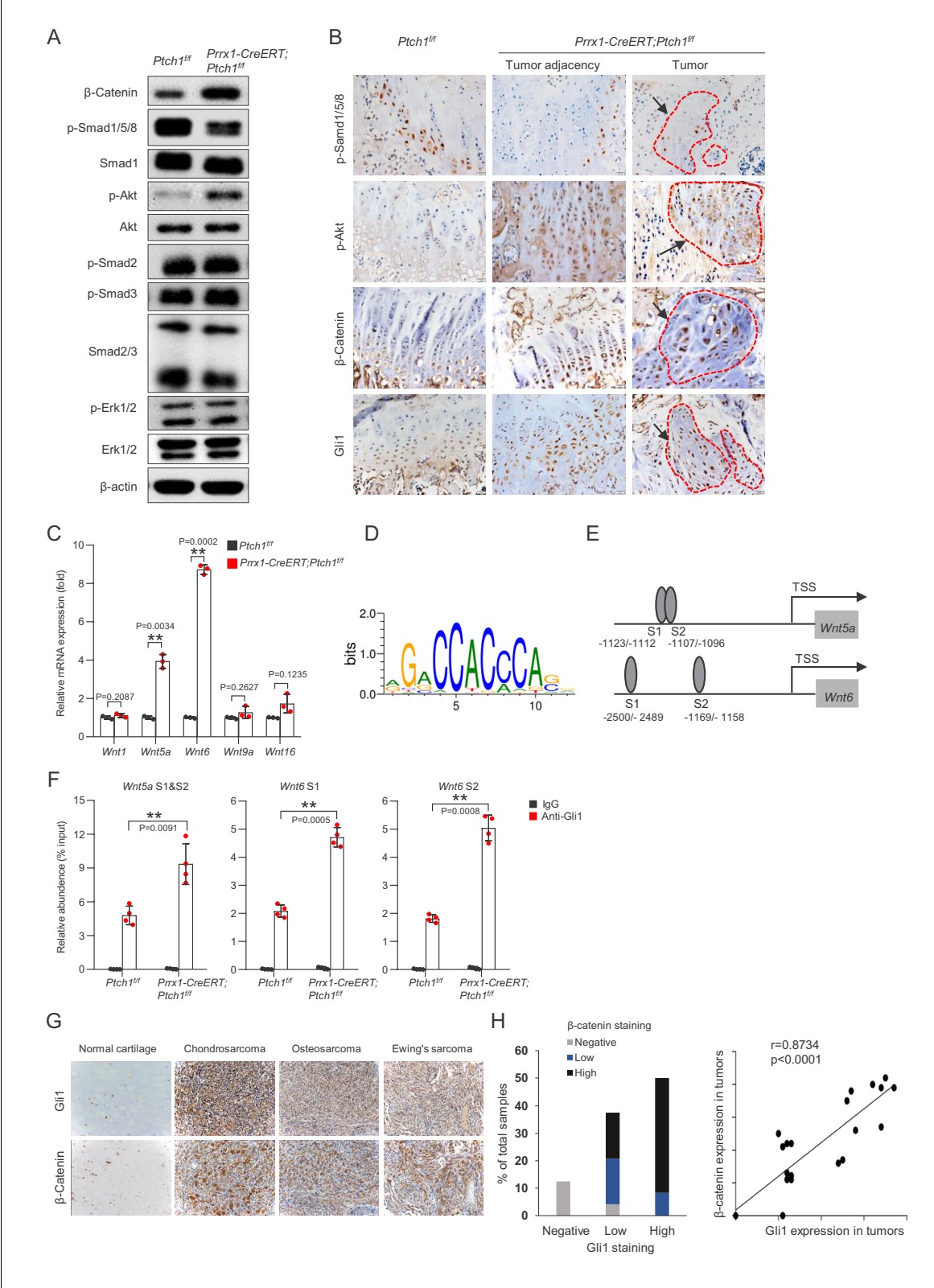

**Figure 7.** Wnt/β-Catenin signaling was activated in $Ptch1^{-/-}$ MSCs and human bone/cartilage tumors. (**A**) Western blot results indicated that MSCs isolated from *Prrx1-CreERT; Ptch1*$^{f/f}$ mice showed enhanced activation of β-Catenin and Akt1 but decreased activation of Smad1/5/8. (**B**) Immunohistochemical staining confirmed that *Prrx1-CreERT; Ptch1*$^{f/f}$ mouse bone section showed increased β-Catenin and p-Akt1 signals but decreased p-Smad1/5/8 signals. Red circle, tumor region. (**C**) Quantitative PCR analysis of Wnt expression in MSCs isolated from *Prrx1-CreERT; Ptch1*$^{f/f}$

*Figure 7 continued on next page*

*Figure 7 continued*

and control mice. All samples were normalized to GAPDH and then to the controls, **p<0.01, n = 3. (D) Sequence logos of 12-mer Gli1 binding motif was shown. (E) Schematic presentation of the putative Gli1-binding sites in the promoter region of mouse *Wnt5a* and *Wnt6*. (F) Quantitative PCR analysis of the immunoprecipitated DNA in separate experiments. **p<0.01 compared to the *Ptch1^{f/f}* anti-Gli1 group. N = 4. (G) Representative images of Gli1 and β-Catenin expression in cartilage tumors by immunohistochemical staining. (H) Relationship between the protein levels of Gli1 and β-Catenin in human cartilage/bone tumors. Left: Bar graph presentation of co-expression between Gli1 and β-Catenin. Right: correlation analysis between Gli1 and β-Catenin, r = 0.8734, p<0.0001.

DOI: https://doi.org/10.7554/eLife.50208.018

The following figure supplements are available for figure 7:

**Figure supplement 1.** SHH or Ptch1 deficiency can activate β-Catenin and increase expression of *Wnt5a* and *Wnt6*.

DOI: https://doi.org/10.7554/eLife.50208.019

**Figure supplement 2.** Images of human cartilage/bone tumor tissue arrays.

DOI: https://doi.org/10.7554/eLife.50208.020

osteosarcomas are derived from Prrx1[+] lineage located at cartilage and periosteal bone, respectively and they express limited levels of markers for differentiated chondrocytes and osteoblasts, respectively. This is in line with the roles of Hh signaling in maintaining cancer stem cells. Although Prrx1[+] MSCs located at different places including the periosteal bones have tri-lineage differentiation potentials, no tumor contains both chondrocytes and osteoblasts in *Prrx1-CreERT; Ptch1^{f/f}* mice. These results, together with previous studies showing that multipotent stem cells must commit to unipotent progenitors in order for *Ptch1* deficiency to promote growth of medulloblastoma (*Schüller et al., 2008*; *Yang et al., 2008b*), suggest that enchondroma and osteosarcoma are originated from the early chondrocyte and osteoblast progenitors derived from the Prrx1[+] MSCs. The underlying reason may be that progenitor cells have much greater proliferation activity than differentiated cells and multipotent stem cells.

The *Prrx1-CreERT; Ptch1^{f/f}* mouse line is one of the few models for cartilage/bone tumors. Other models include mice with chondrocyte-specific Gli-2 overexpression and mice with chondrocyte-specific *Ext1* deletion (*Hirata et al., 2015*; *Hopyan et al., 2002*). While Gli proteins can be activated by Erks, Akt1, and other pro-proliferating signals, Ext1/2 control synthesis of heparin sulphate and mutations of Ext genes may cause IHH diffusion and other effects (*Jones et al., 2010*). In addition, deletion of *Ptch1* in osteoblasts using human osteocalcin-Cre in p53[+/-] but not p53[+/+] background could induce osteosarcoma formation (*Chan et al., 2014*). Here, we show that *Ptch1* deletion in Gli1-marked chondrocytes and osteoprogenitors does not cause tumorigenesis. Thus, *Prrx1-CreERT; Ptch1^{f/f}* mouse represents a unique model useful for dissecting the initiation and progression of both enchondroma and osteosarcoma and for testing drug candidates to target these two tumors.

In addition, we show that Hh signaling promotes MSC proliferation and MSC osteogenic and chondrogenic differentiation but inhibits MSC adipogenic differentiation. While MSC-specific *Ptch1* deletion reproduces some of the phenotypes observed in chondrocyte- and/or osteoblast-specific *Ptch1* knockout mice, for example decreased body weight, body length and/or joint deformation, differences are also evident. *Prrx1-CreERT; Ptch1^{f/f}* mice show unaltered bone mass, whereas *Ptch1* deletion in mature osteoblasts leads to a loss of bone mass due to increased bone resorption (*Mak et al., 2008*). While deletion of *Ptch1* in Col2 chondrocytes results in a delay in hypertrophic growth (*Mak et al., 2006*), we show that deletion of *Ptch1* in MSCs results in enhanced hypertrophic growth. In addition, *Ptch1* deletion in MSCs causes much severe joint deformation and exostoses and only *Ptch1* deletion in MSCs induces cartilage/bone tumor formation. These findings suggest that Hh activation-induced overproliferation and tumorigenesis occur in MSCs but not in differentiated daughter cells. We have previously reported that ectopic expression of HB-EGF or deletion of *Tsc1* (an mTOR inhibitor) in MSCs produces much severe cartilage and bone defects than similar genetic manipulation in chondrocytes and/or osteoblasts (*Li et al., 2019*). Taken together, these results indicate that in the process of osteoblast and chondrocyte production, important regulation may occur at stages of stem cell expansion and commitment.

It is known that Hh signaling promotes cell proliferation, inhibits apoptosis, and promotes tumorigenesis via the canonical and non-canonical pathways. Our current study clearly shows that the canonical pathway plays a critical role in *Ptch1* deficiency-induced enchondroma and osteosarcoma formation, as inhibitors for Smo or Gli1/2, which are developed as candidate drugs to treat cartilage

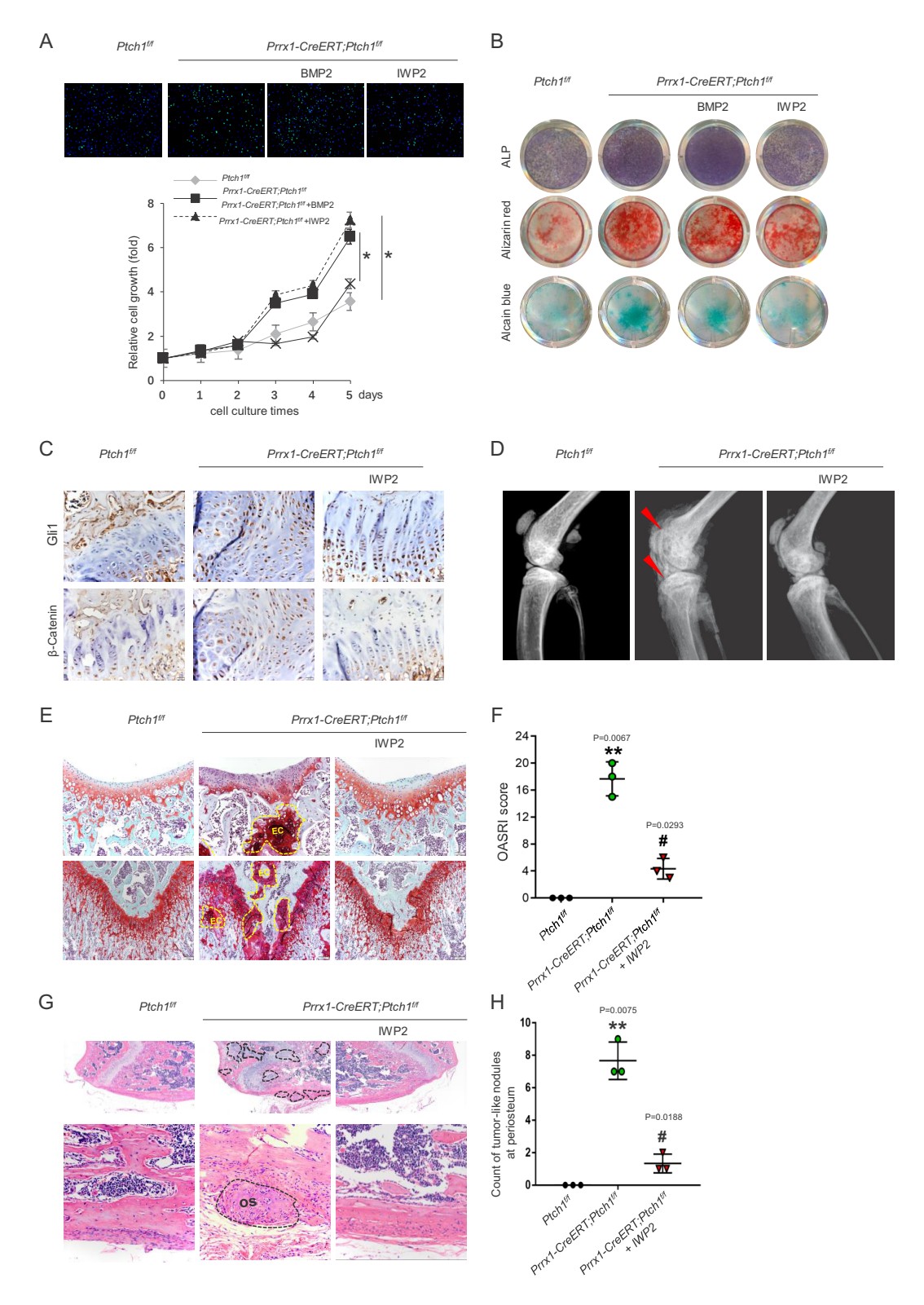

**Figure 8.** Wnt/β-Catenin mediated the pro-tumorigenic activity of *Ptch1* ablation. (**A**) IWP2 suppressed proliferation of BM-MSCs isolated from *Prrx1-CreERT; Ptch1^{f/f}* mouse. Top, Ki67 staining; lower, cck8 assay. *p<0.05. (**B**) IWP2 rescued the enhanced osteogenic and chondrogenic differentiation of BM MSCs isolated from *Prrx1-CreERT; Ptch1^{f/f}* mouse. (**C**) IWP2 inhibited Wnt/β-Catenin signaling without affecting the levels of Gli1. (**D**) Radiographic images showed that IWP2 rescued joint disruption and exostoses in *Prrx1-CreERT; Ptch1^{f/f}* mouse. (**E**) Safranin O staining showed that IWP2 rescued

*Figure 8 continued on next page*

*Figure 8 continued*
joint disruption and cartilage tumor formation in *Prrx1-CreERT; Ptch1^{f/f}* mouse. (F) OARSI scores of control and *Prrx1-CreERT; Ptch1^{f/f}* mice treated with either vehicle or IWP2. \*\*p<0.01 compared to the *Ptch1^{f/f}* group; # p<0.05 compared to the vehicle-treated *Prrx1-CreERT; Ptch1^{f/f}* group, n = 3 per group. (G) H/E-staining showed that IWP2 rescued bone tumor formation in *Prrx1-CreERT; Ptch1^{f/f}* mouse. (H) Osteogenic tumor-like nodules counts in control and *Prrx1-CreERT; Ptch1^{f/f}* mice treated with either vehicle or IWP2. \*\*p<0.01 compared to the *Ptch1^{f/f}* group; #p<0.05 compared to the vehicle-treated *Prrx1-CreERT; Ptch1^{f/f}* group, n = 3 per group.
DOI: https://doi.org/10.7554/eLife.50208.021
The following figure supplements are available for figure 8:
**Figure supplement 1.** FH353 rescued accelerated MSC osteogenic and chondrogenic differentiation caused by *Ptch1* ablation.
DOI: https://doi.org/10.7554/eLife.50208.022
**Figure supplement 2.** IWP2 rescued the anomalies of the paws of *Prrx1-CreERT; Ptch1^{f/f}* mice.
DOI: https://doi.org/10.7554/eLife.50208.023

and bone tumors (*Amakye et al., 2013*), suppressed tumorigenesis. In addition, we find that Wnt5a/6 are up-regulated in-*Ptch1* deficient MSCs and that β-Catenin is highly activated in mouse enchondroma and osteosarcoma samples. Previous studies have shown that canonical Wnt/β-Catenin signaling is essential for skeletal lineage differentiation (*Hill et al., 2005*), and Wnt/β-Catenin signaling is also activated in most of the human cartilage/bone tumor samples, which is correlated with activation of Hh signaling. Functionally, inhibition of Wnt/β-Catenin signaling impedes development of cartilage/bone tumors and other skeletal growth defects. These findings underscore the roles played by the Wnt/β-Catenin pathway in Hh signaling-induced cell proliferation and tumorigenesis and suggest that inhibitors for Wnt/β-Catenin pathway may be useful for treating cartilage/bone tumors, especially the ones that are resistant to Hh signaling inhibitors.

Our findings thus uncover a functional link between Hh signaling and Wnt/β-Catenin signaling in MSC proliferation/differentiation, skeletal growth, and tumor formation. However, previous studies have revealed that in Col2^+ chondrocytes, Wnt/β-Catenin may act upstream of or parallel to IHH pathway in controlling cell survival and joint development (*Mak et al., 2006*). Although Hh pathway-driven development of basal cell carcinoma can be mediated by increased Wnt/β-Catenin signaling (*Yang et al., 2008a*), this is not a common theme as only a limited number of samples display this link (*Adolphe et al., 2006*). Overall, these findings suggest that the functional link between Hh signaling and Wnt/β-Catenin may be specific to MSCs and progenitors of osteoblasts and chondrocytes. This is consistent with the pro-proliferation activity of Wnt/β-Catenin signaling in osteoblastogenesis and chondrogenesis.

In summary, our study has uncovered unanticipated functions of Hh signaling in MSCs, many of which are not observed in the daughter cells of MSCs. Hh activation promotes MSC proliferation and osteoblastic and chondrogenic differentiation but inhibits MSC adipogenic differentiation and leads to development of osteoarthritis, exostoses, and cartilage/bone tumors. Mechanistically, Hh signaling executes its pleiotropic effects via increasing the expression of a couple of Wnt molecules and enhancing Wnt/β-Catenin activation. Moreover, our genetic evidence suggests that enchondroma and osteosarcoma are derived from early progenitors of chondrocytes and osteoblasts and that Wnt/β-Catenin can be targeted to treat cartilage/bone tumors.

## Materials and methods

### Mouse strains

All mouse work was carried out following the recommendations from the National Research Council Guide for the Care and Use of Laboratory Animals, with the protocols approved by the Institutional Animal Care and Use Committee of Shanghai, China [SYXK (SH) 2011–0112]. ROSA-Ai14 (stock #007914), *Gli1-CreERT* (stock #007913), and Floxed *Ptch1* (stock #030494) mouse lines were purchased from The Jackson Laboratory. These mice were kept in the SPF facility of Shanghai Jiao Tong University. *Ptch1^{f/f}* mice were crossed with *Prrx1-CreERT* mice to generate *Prrx1-CreERT; Ptch1^{f/f}* mice. *Prrx1-CreERT; Ptch1^{f/f}* mice were crossed with ROSA-Ai14 mice to generate *Prrx1-CreERT; Ptch1^{f/f};Ai14* mice. *Ptch1^{f/f}* mice were crossed with *Gli1-CreERT* mice to generate *Gli1-CreERT; Ptch1^{f/f}* mice. In all mice, only one allele of *CreERT* is used.

## X-ray and micro-CT analysis

The whole-body and femur radiographs were taken using Cabinet X-Ray system (LX-60, Faxitron Bioptics) with standardized settings (45Kv for 8 s). Quantitative analysis was performed in mouse femur on a SkyScan-1176 micro-CT Scanner (Bruker micro-CT, Belgium), following the procedures provided by the manufacturer. Briefly, scanning was performed using 8.96 μm voxel size, 45KV, 500 μA and 0.6 degrees rotation step (180 degrees angular range) through the whole-length of the femora and extended proximally for 1400 slices. We started morphometric analysis with the first slice, where the femoral condyles were fully merged and extended for 150 slices proximally. Using a contouring tool, we segmented the trabecular bone from the cortical shell manually on key slices, and morphed the contours automatically to segment the trabecular bone on all slices. The three-dimensional structure and morphometry were constructed and analyzed for BV/TV (%), BMD (mg HA/mm$^3$), Tb.N (mm$^{-1}$), Tb.Th (mm) and Tb.Sp (mm).

## Histological analysis

Femur and tibiae bones were fixed in 4% PBS-buffered paraformaldehyde overnight at 4°C. The samples were then stored in 70% alcohol for further experiments. For paraffin sections, samples were decalcified in 15% EDTA for 2 weeks and then dehydrated in alcohol, cleared with xylene, and embedded in paraffin. Four-μm-thick sections were cut using microtome (Lexica Microsystems Nussle GmbH). Hematoxylin and eosin (H/E) or Safranin O staining was carried out on bone sections. Stained slides were photographed under a light microscope (Olympus Microsystems).

## OARSI measurement

Osteoarthritis severity was quantified by the Osteoarthritis Research Society International (OARSI) scoring system (0–6 for grade and 0–24 for score), which was assessed by a single observer who was blinded to the experimental groups. The average thickness of the articular cartilage of the femoral plateau was measured using Image-Pro Plus software.

## Isolation and culture of BM-MSCs and periosteal cells

For BM MSC isolation, mice were euthanized and the femur and tibia were extracted and cleaned. The bone ends were cut off and the bone marrow was flushed out with α-MEM. Single cell suspension was filtered through a 70 μm mesh to remove the debris. BM MSCs were cultured in α-MEM containing 15% FBS, 100 μg/ml penicillin, and 100 μg/ml streptomycin, at 37°C for 5 days. The non-adherent cells were washed out and the BM MSCs were used for further experiments.

For periosteal cell isolation, mice were euthanized and the femur and tibia were extracted and cleaned. The whole bone was digested by dispase and type II collagenase at 37°C for 3 hr without flushing bone marrow out. The digested periosteal tissues were filtered through 70 μm cell strainers and then centrifuged. The cell pellet was resuspended in PBS for FACS sorting. The whole cells or sorted Ai14$^+$ cells were cultured for further studies.

## Colony-forming unit (CFU) assay

BM MSCs were plated at a density of $5 \times 10^6$ cells per well in 12-well plates. After 7 days of culture with medium replaced every 3 days, the cultures were fixed and stained with crystal violet staining solution in methanol for 20 min. The dishes were washed with water and allowed to dry. Colonies were counted macroscopically, and data were reported as colony numbers per well.

## Cell counting

Cell proliferation was assessed using the Cell Counting Kit-8 (Sangon Biotech Co., Ltd., Shanghai, China). Briefly, cells were seeded into duplicate wells of 96-well plates at a density of $1 \times 10^3$ cells/well. At days 1, 2, 3, 4, and 5, 10 μl of cck-8 solution was added to each well. The samples were incubated for 4 hr at 37°C. The absorbance of each well was determined at 450 nm. Three independent experiments were performed.

## Cell migration assay

Cells were pretreated with mitomycin C (4 mg/ml, Sigma-Aldrich) for 2 hr prior to analysis. Cell migration was monitored at 0, 10, 20 hr by introduction of a scratch in confluent cells.

## Differentiation of MSCs

To induce BM MSC tri-lineage differentiation, sorted Prrx1[+] lineage BM MSCs were seeded at $5 \times 10^4$/well in 12-well plates. The next day, the cells were switched into osteogenic medium with α-MEM medium containing 15% FBS, 10 mM β-glycerol phosphate, and 50 µg/ml ascorbic acid, for 7–10 days, with medium changed every 2 days. The cells were then fixed in 4% paraformaldehyde and stained for ALP using an Alkaline Phosphatase Kit (Sigma-Aldrich) or Alizarin red. For mineralization assay, the cells were cultured for 21 days, which were stained in 5% silver nitrate solution under ultraviolet light or 1% alizarin red S. The silver staining was terminated by adding sodium thiosulfate solution.

For adipocyte differentiation, BM MSCs were plated at $1 \times 10^5$/well in a 12-well plate and cultured in α-MEM containing 15% FBS, 100 nM dexamethasone, and 5 µM insulin for 2 weeks. The cells were then fixed and stained with Oil red O solution.

For chondrogenesis assays, BM MSCs were suspended at a concentration of $1.6 \times 10^7$ cells/ml. We generated micro-mass cultures by seeding 10 µl droplets of cell suspension at the center of 12-well plates. Cells were allowed to attach for 2 hr before adding α-MEM containing 15% FBS, 100 IU/ml penicillin, and 100 µg/ml streptomycin. After 1 day, the medium was replaced with chondrogenic medium in α-MEM containing 15% FBS, 100 nM dexamethasone, 10 ng/ml TGFβ1, and 1 µM ascorbate-2-phosphate. Cultures were maintained for 21 days, with the medium changed every 3 days, and were lastly stained with Alcian Blue or Toluidine blue.

## In vivo ectopic bone formation assay

A total of $2 \times 10^6$ isolated BM MSCs were collected and incubated with 40 mg hydroxyapatite/tricalcium phosphate carrier (HA/TCP: 12.5:87.5 by weight) (Bioengineering Research Center of Sichuan University, China) scaffolds for 6 hr at 37°C in humidifying incubator, and then implanted subcutaneously onto the back of 2-month-old BALB/C homozygous nude (nu/nu) mice (four mice per group). Mice were euthanized 10 weeks later after transplantation. The implants were fixed in 4% paraformaldehyde and then decalcified for 10 days. The sections were stained with H/E. To quantify the bone-like tissues, 10 images of each sample were taken randomly to measure the area of new bone formation versus total area.

## Pellet culture assays (chondrogenesis)

BM MSCs were suspended in chondrogenic medium consisting of high-glucose DMEM supplemented with 10 ng/ml recombinant human transforming growth factor-β3 (TGF-β3; R&D), 100 nM dexamethasone (Sigma), 50 µg L-ascorbic acid/ml (Sigma), 1 mM sodium pyruvate, 40 µg proline/ml and ITS + premix (Sigma; final concentrations: 6.25 µg/ml bovine insulin, 6.25 µg/ml transferrin, 6.25 µg/ml selenous acid, 5.33 µg/ml linoleic acid and 1.25 mg/ml bovine serum albumin). Aliquots of $5 \times 10^5$ cells, suspended in 500 µl chondrogenic medium, were centrifuged at 300 g for 5 min in 15 ml polypropylene conical tubes. Pelleted cells were incubated at 37°C under 5% $CO_2$ with loosened caps to permit gas exchange. Within 24 hr of incubation, the sedimented cells formed a spherical aggregate at the bottom of the tube. The medium was changed every 3 days and pellets were harvested after 6 weeks.

## Quantitative PCR

Total RNA was extracted using Trizol regent (Invitrogen), which was reverse transcribed using Transcriptor Universal cDNA Master (Roche) following the manufacturer's instructions. Quantitative PCR was carried out using Fast Start Universal SYBR Green Master kit (Roche) on ABI Prism 7500 Sequence Detection System (Applied Biosystems) using primers listed in *Supplementary file 1* Table S1. The levels of different mRNA species were calculated with the delta-delta CT method and normalized to GAPDH. Significant difference was analyzed using two-tailed Student's t-test.

## Immunofluorescence and immunohistochemical staining

Decalcified bone sections were deparaffinized in xylene, rehydrated in ethanol, and permeabilized with 0.1% Triton X-100 for 20 min at room temperature. Antigen retrieval was performed in a citrate buffer at 95°C for 20 min. For immunostaining, the bone slides or cells on cover glass were blocked with 10% goat serum for 60 min, incubated with primary antibodies overnight at 4°C, washed in PBS,

incubated with secondary antibody for 1 hr at 37°C, and washed with PBS before mounted on Pro-Long Gold DAPI (Life Technologies). The antibodies used in this study were: Col1α (ab21286, Abcam), Col2 (ab34712, Abcam), Col10 (ab58632, Abcam), CD31(550274, BD), Vimentin (ab92547, Abcam), Perilipin (ab3526, Abcam), Non-immune immunoglobulin G (IgG) (of the same species as the primary antibodies) used as negative control. The secondary antibodies were goat anti-rabbit Alexa Fluor 488 (ThermoFisher Scientific). Slides were mounted with antifade mounting medium with DAPI (ThermoFisher Scientific). Cell proliferation was determined by Ki67 immunofluorescence staining (Abcam, ab15580). Images were taken under Olympus DP72 microscope (Olympus Microsystems).

For immunohistochemical staining, endogenous peroxidase activity was quenched with 3% $H_2O_2$ in methanol for 20 mins followed by washing with PBS before primary antibody incubation. After incubation with secondary antibody, sections were developed with diaminobenzidine and counterstained with hematoxylin and then dehydrated and mounted in neutral resins. Primary antibodies were: Ki67 (ab15580, Abcam), p-Akt1 (4060S, CST), p-Smad1/5/8 (13820S, CST), FSP1 (ab27957, Abcam), Gli1(43926, SAB), β-Catenin (ab32572, Abcam), Non-immune immunoglobulin G (IgG) (of the same species as the primary antibodies) was used as negative controls.

## Human bone tumor tissue arrays

The human bone tumor tissue chips were provided by Alenabio (Xi'an, China). Each tissue chip was included 25 samples of human bone tumor with 10 chondrosarcoma, 8 osteosarcoma and 6 Ewing's sarcoma and one normal cartilage as control. The chips were stained with Gli1 ((43926, SAB) or β-Catenin ((ab32572, Abcam) antibodies under standard IHC protocol. The stained slides were examined under Olympus DP72 microscope (Olympus Microsystems), and images were acquired.

## Western blot

Total proteins were extracted from cells or tissues with TNEN buffer containing phosphatase and proteinase inhibitors, quantitated by the Bradford method (Bio-Rad assay), and subjected to SDS-PAGE gel electrophoresis, which were transferred onto nitrocellulose membranes. The proteins were detected with specific antibodies using standard western blot method. The following antibodies were used: β-Actin (sc-47778, Santa Cruz), p-Akt1 (9271S, CST), Akt1 (9272S, CST), p-Erk1/2 (4377, CST), Erk1/2 (9102S, CST), p-Smad1/5/8 (9511L, CST), Smad1 (9743L, CST), Ptch1 (17520, Proteintech), Gli1 (2553S, CST), p-Smad2 (3101S, CST), p-Smad3 (9520S, CST), Smad2/3 (3102, CST), β-Catenin (ab32572, Abcam). Immunoreactivity was detected using a Western Chemiluminescent HRP Substrate Kit (Millipore) and imaged with FluorChem M system (Protein Simple).

## SHH and inhibitor administration

For in vitro experiments, the IWP2 (Selleck Chemicals, USA) and FH535 (Selleck Chemicals, USA) were dissolved in dimethyl sulfoxide and applied at a final concentration of 10 µM and 20 µM in cell culture medium, respectively. Recombinant human sonic hedgehog (R and D systems, USA) was dissolved in PBS and applied at a final concentration of 5 µg/ml in cell culture medium.

For in vivo experiments, Cyclopamine (Selleck Chemicals, USA), GANT61 (Selleck Chemicals, USA), and IWP2 were reconstituted in corn oil. Mice were treated with 20 mg/kg cyclopamine, 30 mg/kg GANT61, or 10 mg/kg IWP2 through intragastric administration every other day after tamoxifen injection. Mice were treated for 2 months and then evaluated.

## ChIP assay

The chromatin immunoprecipitation (ChIP) assay was carried out following the manufacturer's protocol (SimpleChIP Enzymatic Chromatin IP Kit, Agarose Beads, #9002). Briefly, BMSCs were crosslinked with 1% formaldehyde and blocked with glycine, which were then washed and digested by micrococcal nuclease. The nuclear pellet was suspended in ChIP buffer and sheared by sonication. An aliquot of each sheared chromatin sample was set aside as input control. The remained chromatin was then incubated with anti-Gli1 antibody (Santa Cruz). Rabbit immunoglobulin G (IgG, #9002, CST) was used as a control. The immunoprecipitated chromatins were then eluted with ChIP elution buffer. The DNA fragments were then released by treatment with ribonuclease A and then with proteinase K at 65°C. The released DNA fragments were purified with columns and amplified by site-

specific primers by Real-time Quantitative PCR assay. The data were analyzed by the following formula: percent (%) input recovery = (100/(input fold dilution/bound fold dilution))×2(input CT−bound CT). Pairs of primers designed to amplify the specific target sequences of the putative promoters were listed in *Supplementary file 1* Table S2.

## Nuclear and cytoplasmic fractionation

Cells were washed with ice-cold PBS and scraped from the wells with the plates on ice. A nuclear and cytoplasmic protein extraction kit (Beyotime, china) was applied to separate these two cellular components following the manufacturer's instructions. The immunoblotting procedure was performed as described before and the following antibodies were used: GAPDH (T40004S, abmart), H3 (ap50907, abgent), and β-Catenin (ab32572, Abcam) antibodies.

## Virus production and infection

To overexpress Ptch1 in BMSCs, PCR-amplified full-length human Ptch1 cDNA was tagged with Flag and subcloned into the pHBLV-CMV-Puro vector (Hanbio Biotechnology, China). Lentiviral vector carrying GFP was constructed as negative control. Recombinant viruses were collected and purified and titer was determined. For lentivirus infections, BMSCs were plated at a density of $1 \times 10^5$ cells in 12 well-plate. LV-GFP, LV-hPtch1, or LV-Cre viruses were added at a multiplicity of infection of 20 when the cells were 30–50% confluent. After 72 hr, the transfected cells were harvested and gene overexpression was verified by Western blotting. For transient *Ptch1* deletion, primary *Ptch1^{f/f}* MSCs were infected with Cre-expressing lentivirus.

## Statistical analyses

Numerical data and histograms were expressed as the mean ± SD. Comparisons between two groups were analyzed using two-tailed unpaired Student's t test. $p < 0.05$ was considered statistically significant. Analysis of mice was litter-based and at least three litters were analyzed for every parameter. All the experiments were repeated at least three times.

## Acknowledgements

We thank Dr. Shunichi Murakami for providing *Prrx1-CreERT* mice. The work was supported by the National Key Research and Development Program of China (2017YFA0103602), the National Natural Science Foundation of China (81373210, 81520108012 and 91542120), and the Schaefer Research Scholarship to BL.

## Additional information

### Funding

| Funder | Grant reference number | Author |
| --- | --- | --- |
| National Key Research and Development Program of China | 2017YFA0103602 | Huijuan Liu |
| National Key Research and Development Program of China | 2018YFA0800803 | Baojie Li |
| National Natural Science Foundation of China | 81520108012 | Baojie Li |
| National Natural Science Foundation of China | 91542120 | Baojie Li |
| Columbia University | Schaefer Research Scholarship | Baojie Li |

The funders had no role in study design, data collection and interpretation, or the decision to submit the work for publication.

## Author contributions
Qi Deng, Data curation, Formal analysis, Investigation, Methodology, Project administration; Ping Li, Investigation, Project administration; Manju Che, Jiajia Liu, Soma Biswas, Investigation; Gang Ma, Lin He, Resources; Zhanying Wei, Software; Zhenlin Zhang, Resources, Software; Yingzi Yang, Writing—review and editing; Huijuan Liu, Project administration; Baojie Li, Conceptualization, Data curation, Formal analysis, Funding acquisition

## Author ORCIDs
Soma Biswas (iD) http://orcid.org/0000-0002-1427-2678
Yingzi Yang (iD) http://orcid.org/0000-0003-3933-887X
Baojie Li (iD) https://orcid.org/0000-0002-3913-1062

## Ethics
Animal experimentation: All mouse work was carried out following the recommendations from the National Research Council Guide for the Care and Use of Laboratory Animals, with the protocols approved by the Institutional Animal Care and Use Committee of Shanghai, China [SYXK (SH) 2011-0112]. All surgery was performed under sodium pentobarbital anesthesia, and every effort was made to minimize suffering.

## Decision letter and Author response
Decision letter https://doi.org/10.7554/eLife.50208.027
Author response https://doi.org/10.7554/eLife.50208.028

## Additional files

### Supplementary files
• Supplementary file 1. Key resources table.
DOI: https://doi.org/10.7554/eLife.50208.024

• Supplementary file 2. The primers for qPCR analysis.
DOI: https://doi.org/10.7554/eLife.50208.025

### Data availability
All data generated or analysed during this study are included in the manuscript and supporting files.

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
