## [Decision Letter]

[Editors’ note: a previous version of this study was rejected after peer review, but the authors submitted for reconsideration. The first decision letter after peer review is shown below.]

Thank you for choosing to send your work, "Activation of Hedgehog signaling in mesenchymal stem cells induces cartilage and bone tumor formation via Wnt/β-Catenin", for consideration at *eLife*. Your article has been reviewed by three peer reviewers, and the evaluation has been overseen by a Reviewing/Senior Editor. The following individual involved in review of your submission has agreed to reveal their identity: Danny Chan (Reviewer #2).

Although the work is of interest, we regret to inform you that the findings at this stage are too preliminary for further consideration at *eLife*.

Specifically, the reviewers felt that this was a potentially interesting manuscript that suggests a link between hedgehog and Wnt signaling in bone tumor biology. However, the reviewers noted that more direct links and further experimentation would be needed to support the claims. For example, there is a lack of a clear study of the initial pool of cells labeled in the cell tracing experiment and the link between HH and Wnt signaling is suggestive but not definitive. I refer you to the full reviews below for clarification. If you feel you can address the reviewers' concerns at some later time, we would be happy to consider a new paper on this topic that includes a rebuttal to the reviews. In that event, every effort would be made to send the manuscript back to the same reviewers.

*Reviewer #1:*

The authors show in this study that deletion of the inhibitor of the Indian Hedgehog (IHH) signaling pathway Patched 1 (Ptch1) in mesenchymal progenitor cells using *Prrx1-Cre* driver mice promotes osteoblast and chondrocyte differentiation and results in an osteoarthritis phenotype in 3 month old mice and appearance of tumors. The authors further try to link the appearance of these tumors to activation of the Wnt/β-Catenin pathway. This is potentially an important study and a red flag given the interest in Wnt signaling as a cure for otherwise milder diseases of the skeleton; however, this study needs to be fleshed out more to make a convincing argument.

- The notion that Wnt signaling during development is need for chondrocyte differentiation is not known and was shown 15 years ago by one of the authors and by the Hartmann group, this should be acknowledged in a clearer manner.

- The study of the mouse models is for now quite superficial: on what histological, molecular or phenotypic bases do the authors speak of osteosarcoma? When do these tumors appear? Do they metastasize as osteosarcoma do? How long do the mice live? This part of the study is sketchy and needs to expanded.

- Can the authors link genetically IHH signaling and Wnt signaling in this cell type?

- The authors show a clear increase in β-Catenin but they do not show whether it is cytoplasmic or nuclear.

*Reviewer #2:*

This is a potentially interesting manuscript to provide a link between hedgehog (HH) and Wnt signaling in bone tumor biology. Both HH and Wnt signals have implicated in bone tumor formation such as osteosarcoma and enchondroma; thus their involvement is not novel. This manuscript suggests that by inactivation Ptch1 in postnatal (P14) bone marrow MSCs using the *Prrx1-CreERT* in mice, this would induce a constitutive activation of the Smo pathways in these cells, that subsequently contributes to a high incidence for the formation of osteosarcoma and enchondroma. They showed a correlation between activation of the HH pathway to Wnt, and propose that in this context of the MSCs, HH act upstream of Wnt. However, they further propose that osteosarcoma and enchondroma are formed from distinct progenitor lineages and not from a common multi-potential MSC pool. In support of their hypothesis, they provided evidence for the direct involvement of the HH pathway by inhibiting SMO and Gli2 in mice to suppress the formation of osteosarcoma and enchondroma, as with the inhibition of the Wnt pathway in mice.

While the data is interesting and supportive through association, more direct links are needed to support their claims. A major concern for me is the lack of a clear study of the initial pool of cells that were labeled in the cell tracing experiment, and thus the cells in which *Ptch1* are inactivated. They must show a thorough analysis of the cells in the axial and appendicular bones/cartilage and bone marrow immediately after the 4 days of TAM injection, and not 2.5 months later. Further, an elected time course study after TAM injection till 2.5 months would be informative and necessary as to the proliferative and differentiation of cells. It seem odd to me that the periosteal cells are labeled if Prrx1 only target MSCs. If it is indeed a more differentiated or committee pool of MSCs for the skeletal lineage, this would be interesting but need to be established.

The link between HH and Wnt is supportive, but not definitive. In development, the relationship between these two pathways is complex and context dependent. They propose that the function of HH in MSCs and chondrocytes, but this could just be the context that they are different cells, and therefore, the downstream outcomes are different.

Based on the concerns above, there is substantial correction and additional data required for further consideration.

In addition, while in general, the figures and data are of good quality, the presentation needs to be more stringent to provide the basic information for readers to follow and understand the key points. Generally, there is a lack of details in the description and labeling of the figures, with readers having to second guess what samples and what region(s) of the skeletal elements are being described. Figure 3—figure supplement 2 is missing.

*Reviewer #3:*

The paper entitled "Activation of Hedgehog signaling in mesenchymal stem cells induces cartilage and bone tumor formation via the Wnt/β-Catenin pathway" is potentially interesting. The authors showed that *Ptch1* deletion led to development of osteoarthritis-like phenotypes, exostoses, enchondroma, and osteosarcoma in Smo-Gli1/2-dependent manners. *Ptch1* deletion increases the expression of *Wnt5a/6* and leads to enhanced β-Catenin activation. Inhibiting Wnt/β-Catenin pathway suppresses the development of skeletal anomalies, enchondroma, and osteosarcoma. The study is well designed and presented.

There are several issues:

How does *Ptch1* overexpression or knockdown affect osteoblasts and condroctytes?

Figure 1: It looks like that the bone density in *Ptch1* deletion is weaker?

Figure 2: It will be informative to measure cartilage thickness and subchondrol bone density as key OA parameters.

Figure 2: Have OARSI Osteoarthritis Cartilage Histopathology Assessment scores been measured?

Figure 3: When osteosarcoma was observed? And what percentage of mice? Will this lead to dead of the mice at later stage? Will overexpression of *Ptch1* rescue this phenotype?

Figure 4: Any quantitative measurement like OARSI scoring?

Figure 5E, F: Any quantitative measurement?

Figure 7E, F: Any quantitative measurement?

Figure 2—figure supplement 1: Any quantitative measurement?

[Editors’ note: what now follows is the decision letter after the authors submitted for further consideration.]

Thank you for submitting your article "Activation of Hedgehog signaling in mesenchymal stem cells induces cartilage and bone tumor formation via Wnt/β-Catenin" for consideration by *eLife*. Your article has been reviewed by three peer reviewers, and the evaluation has been overseen Marianne Bronner as the Senior and Reviewing Editor. The following individual involved in review of your submission has agreed to reveal their identity: Danny Chan (Reviewer #2).

The reviewers have discussed the reviews with one another and the Reviewing Editor has drafted this decision to help you prepare a revised submission.

Essential Revisions:

The reviewers are mostly happy with your revision but ask you to make two essential changes:

1) Improve their list of references, which is still self-referential and full of review articles instead of original articles.

2) Resolve the issue in the cell tracing data from P14 as there was high autofluorescence, masking other potential location of *Prxx1* expressing cells. At the very least, it would be important to acknowledge the limitations in the Discussion.

Full reviews are attached below for further details.

Reviewer #1:

The authors have addressed my questions and the manuscript has been improved.

Reviewer #2:

This revision has largely addressed my concerns. The authors have provided additional data and clarified many of the issues. I only have a few comments for the authors to consider that I feel could further clarify the nature and source of the *Prrx1* expressing cells at P14 contributing to the formation of endochondroma and osteosarcoma.

Some concern still related to the data on the source of the "MSCs" labeled at P14 using the *Prxx1-Cre* mouse shown in Figure 5. While I appreciate the author's effort in addressing this critical issue, the autofluorescence signal makes it hard to delineate the precise location of cells that were labeled after the 4 consecutive injections with TAM. There is a need to show a control with vehicle injections to assess the level of autofluorescence. Further, the experiments for the differentiation of *Prxx1^+^* bone marrow MSCs and its correlation of the "MSCs" within the cartilage and periosteal bone is difficult to appreciate, as the context is very different. Can the author provide some assessment/comment on whether any of the marrow MSCs can contribute to the enchondromas as they have not eliminated such a possibility.

The overall finding is very interesting. In the Discussion, the authors have provide what we already know concerning the role of HH and Wnt signaling in skeletogenesis; can the authors comment on what could be the differences in the action of HH when it is activated in "osteochondral progenitor cells" as oppose to differentiated osteoblasts and chondrocytes with respective to the relationship between HH and Wnt signaling leading the formation of tumors.

Reviewer #3:

The authors have addressed most of my experimental comments, which improves the paper. What remains and should be addressed for a high visibility paper is an extreme reliance on (1) review articles (for instance, 7 out of the first 16 references are review articles and it does not improve after) and (2) on citations of their own work or of their collaborators. This would be perfectly acceptable if they had done the original work but this is not the case. This needs to be addressed before going further.

---

## [Author Response]

[Editors’ note: the author responses to the first round of peer review follow.]

Reviewer #1:The authors show in this study that deletion of the inhibitor of the Indian Hedgehog (IHH) signaling pathway patched 1 (Ptch1) in mesenchymal progenitor cells using Prrx1-Cre driver mice promotes osteoblast and chondrocyte differentiation and results in an osteoarthritis phenotype in 3 month old mice and appearance of tumors. The authors further try to link the appearance of these tumors to activation of the Wnt/β-Catenin pathway. This is potentially an important study and a red flag given the interest in Wnt signaling as a cure for otherwise milder diseases of the skeleton; however, this study needs to be fleshed out more to make a convincing argument.- The notion that Wnt signaling during development is need for chondrocyte differentiation is not known and was shown 15 years ago by one of the authors and by the Hartmann group, this should be acknowledged in a clearer manner.

Thanks for pointing this out. We have cited these papers and described these findings in the new manuscript.

- The study of the mouse models is for now quite superficial: on what histological, molecular or phenotypic bases do the authors speak of osteosarcoma? When do these tumors appear? Do they metastasize as osteosarcoma do? How long do the mice live? This part of the study is sketchy and needs to expanded.

We agree with the reviewer. We have now added detailed description about osteosarcoma in the first paragraph of the subsection “Ptch1 deficiency led to osteosarcoma formation at periosteal surfaces”.

In particular, the appearance and unique locations of the tumors, the histology of the tumors, the expression of osteoblast-specific but not chondrocyte-specific markers, and pulmonary metastasis (2 out of 9 mice) (new Figure 3—figure supplement 1D), support that these tumors are osteosarcoma. We also presented the time course analysis of enchondroma and osteosarcoma to show that these tumors developed early at day 14 for enchondroma and day 30 for osteosarcoma-like mass after induction of *Ptch1* deletion (new Figure 5A, Figure 2—figure supplement 2A, and Figure 3—figure supplement 1A). The mice can survive up to 7 months after TAM injection. We have included the information in the new manuscript.

- Can the authors link genetically IHH signaling and Wnt signaling in this cell type?

We have already shown that *Ptch1^-/-^* BM-MSCs displayed an increase in the β-Catenin protein and increases in mRNA levels of *Wnt5a* and *Wnt6* (Figure 7A and 7C). In addition, we treated MSCs with recombinant SHH and found that this led to activation of β-Catenin and increased expression of *Wnt5* and *Wnt6a*. However, SHH failed to further increase *Wnt5a* and *Wnt6* expression in *Ptch1^-/-^* BM-MSCs (new Figure 7—figure supplement 1B and 1C). Lastly, using *Ptch1^f/f^* BM-MSCs, we expressed Cre using viruses and found that transient deletion of Ptch1 led to activation of β-Catenin in the nucleus (new Figure 7—figure supplement 1A). These results established the link between IHH and Wnt signaling in BM-MSCs.

- The authors show a clear increase in β-Catenin but they do not show whether it is cytoplasmic or nuclear.

To address this point, we have fractionated the nuclear and cytoplasmic proteins from BM-MSCs and western blot results showed that there was an increase in both cytoplasmic and nuclear β-Catenin, supporting that the pathway was activated (new Figure 7—figure supplement 1A).

Reviewer #2:[…] While the data is interesting and supportive through association, more direct links are needed to support their claims. A major concern for me is the lack of a clear study of the initial pool of cells that were labeled in the cell tracing experiment, and thus the cells in which Ptch1 are inactivated. They must show a thorough analysis of the cells in the axial and appendicular bones/cartilage and bone marrow immediately after the 4 days of TAM injection, and not 2.5 months later. Further, an elected time course study after TAM injection till 2.5 months would be informative and necessary as to the proliferative and differentiation of cells. It seem odd to me that the periosteal cells are labeled if Prrx1 only target MSCs. If it is indeed a more differentiated or committee pool of MSCs for the skeletal lineage, this would be interesting but need to be established.

We traced the Prrx1 lineage cells at different time points after injecting Tamoxifen to 2 week-old *Prrx1-CreERT; tdTomato* mice. We found that Prrx1 could mark cells at the growth plate, endosteal bones, periosteal bones, and bone marrow in long bones (Figure 5A). However, it labelled much fewer osteoblasts and chondrocytes in the vertebrae (Figure 5—figure supplement 1B), suggesting that vertebral bones have different origins from long bones. This is consistent with our observation that deletion of *Ptch1* in *Prrx1^+^* cells did not affect vertebrae bone mass and cartilage or develop tumors (Figure 2—figure supplement 2). Moreover, the time course analysis showed that tumors developed early at day 14 for enchondroma and day 30 for osteosarcoma-like mass after induction of *Ptch1* deletion (Figure 5A, Figure 2—figure supplement 2A, and Figure 3—figure supplement 1A).

Recent studies have identified SSCs (MSCs) in the growth plate genetically marked by PthrP, periosteal SSCs marked by Cathepsin K, and perisinusoidal MSCs marked by Gremlin 1, Nestin, and LepR. SSCs isolated from these places have the potential to differentiate into osteoblasts, chondrocytes, and adipocytes. While previous studies have shown that Prrx1-marked BM-MSCs have tri-lineage differentiation potentials, we found that periosteal *Prrx1^+^* cells also have osteoblast, chondrocyte, and adipocyte differentiation potentials (Figure 5—figure supplement 2B-D). We further showed Prrx1-marked BM-MSCs account for more than 50% of adherent cells (Figure 5—figure supplement 2A). These results suggest that *Prrx1^+^* cells isolated from BM and periosteal bones are MSCs with multi-lineage potencies. Since deletion of *Ptch1* in mature osteoblasts and chondrocytes (including Gli1^+^ osteoprogenitors and chondrocytes) does not cause tumorigenesis (Figure 3—figure supplement 3), we conclude that it is the early progenitors that are transformed. Moreover, even after becoming tumor cells, they continue to progress into the specific lineage. We conclude that Prrx1 may mark cells at several places in the skeleton that have tri-lineage differentiation potentials.

The link between HH and Wnt is supportive, but not definitive. In development, the relationship between these two pathways is complex and context dependent. They propose that the function of HH in MSCs and chondrocytes, but this could just be the context that they are different cells, and therefore, the downstream outcomes are different.

We agree that the crosstalk between HH and Wnt pathways can be affected by contexts. However, our ChIP experiments revealed that the promoters of Wnt5a and Wnt6 contain 2 bonding sites for Gli1, suggesting a direct link (Figure 7D-F). Since the promoters usually have binding sites for other transcription factors, the HH-Gli1-Wnt pathway can still be affected by cell context.

Based on the concerns above, there is substantial correction and additional data required for further consideration.

We have done required experiments and included the new results in the new manuscript.

In addition, while in general, the figures and data are of good quality, the presentation needs to be more stringent to provide the basic information for readers to follow and understand the key points. Generally, there is a lack of details in the description and labeling of the figures, with readers having to second guess what samples and what region(s) of the skeletal elements are being described. Figure 3—figure supplement 2 is missing.

We have added more labeling and arrows in the figures, expanded the legend, and also put back the missing figure.

Reviewer #3:The paper entitled "Activation of Hedgehog signaling in mesenchymal stem cells induces cartilage and bone tumor formation via the Wnt/β-Catenin pathway" is potentially interesting. The authors showed that Ptch1 deletion led to development of osteoarthritis-like phenotypes, exostoses, enchondroma, and osteosarcoma in Smo-Gli1/2-dependent manners. Ptch1 deletion increases the expression of Wnt5a/6 and leads to enhanced β-Catenin activation. Inhibiting Wnt/β-Catenin pathway suppresses the development of skeletal anomalies, enchondroma, and osteosarcoma. The study is well designed and presented.There are several issues:How does Ptch1 overexpression or knockdown affect osteoblasts and condroctytes?

We overexpressed *Ptch1* in wildtype MSCs by virus and found that this inhibited osteoblast and chondrocyte differentiation (new Figure 6—figure supplement 1).

Many of the BM-MSC differentiation data were done using *Ptch1^f/f^* MSCs, which were infected with Cre-expressing viruses, leading to partial deletion of Ptch1. We found that transient deletion of Ptch1 produced similar results as BM-MSCs isolated from *Prrx1-CreERT;Ptch1^f/f^* mice (TAM injected). These transient deletion cells should be similar to knock-down cells. We describe this in the new submission (new Figure 7—figure supplement 1A).

Figure 1: It looks like that the bone density in Ptch1 deletion is weaker?

We performed micro-CT analysis in 4 pairs of mice and found that the bone density showed no significant change in *Ptch1* deletion mice (Figure 3—figure supplement 2). We have now used a more representative image.

Figure 2: It will be informative to measure cartilage thickness and subchondrol bone density as key OA parameters.

We have measured the cartilage thickness by Image pro Plus software based on H/E staining images and subchondral bone density by micro-CT. The results are presented in new Figure 2—figure supplement 1B-D, and described in the new manuscript.

Figure 2: Have OARSI Osteoarthritis Cartilage Histopathology Assessment scores been measured?

We have measured the Osteoarthritis Cartilage Histopathology Assessment scores according to Safranin O staining based on OARSI system (0–6 for grade and 0–24 for score). The results are presented in new Figure 2B and described in the new manuscript, which are indicative of early OA.

Figure 3: When osteosarcoma was observed? And what percentage of mice? Will this lead to dead of the mice at later stage? Will overexpression of Ptch1 rescue this phenotype?

The tumor-like mass was observed in *Prrx1-CreERT;Ptch1^f/f^* 30 days after TAM injection. All 9 mice developed megascopic osteosarcoma. These mice can survive up to 7 months after TAM injection. We have added the description in new manuscript.

Generation of Ptch1 overexpression mice will take a long time. In addition, overexpression of Ptch1, similar to deletion of Smoothened, may cause embryo survival defects. Nevertheless, we used 2 inhibitors of Hh signaling and found these phenotypes are rescued. This should be sufficient to prove the point.

Figure 4: Any quantitative measurement like OARSI scoring?

We have added the OARSI scores in new Figure 4C.

Figure 5E, F: Any quantitative measurement?

We have added quantitative measurement of bone formation rate, adipocytes ratio and chondrogenic pellet sizes, respectively in new Figure 6E and 6F.

Figure 7E, F: Any quantitative measurement?

We have added quantitative data on the numbers of osteogenic tumor-like nodules and OARSI scores in new Figure 8F and 8H.

Figure 2—figure supplement 1: Any quantitative measurement?

We have added statistical results of tumor counts in new Figure 2—figure supplement 2F.

[Editors' note: the author responses to the re-review follow.]

Essential Revisions:The reviewers are mostly happy with your revision but ask you to make two essential changes:1) Improve their list of references, which is still self-referential and full of review articles instead of original articles.

We have removed 3 references authored by us and 13 other review article references.

2) Resolve the issue in the cell tracing data from P14 as there was high autofluorescence, masking other potential location of Prxx1 expressing cells. At the very least, it would be important to acknowledge the limitations in the Discussion.

We have included a control image to show that autofluorescence is low (Figure 5—figure supplement 1A).

Full reviews are attached below for further details.Reviewer #2:This revision has largely addressed my concerns. The authors have provided additional data and clarified many of the issues. I only have a few comments for the authors to consider that I feel could further clarify the nature and source of the Prrx1 expressing cells at P14 contributing to the formation of endochondroma and osteosarcoma.

We have included a control image to show that autofluorescence is low (Figure 5—figure supplement 1A).

Some concern still related to the data on the source of the "MSCs" labeled at P14 using the Prxx1-Cre mouse shown in Figure 5. While I appreciate the author's effort in addressing this critical issue, the autofluorescence signal makes it hard to delineate the precise location of cells that were labeled after the 4 consecutive injections with TAM. There is a need to show a control with vehicle injections to assess the level of autofluorescence. Further, the experiments for the differentiation of Prxx1^+^ bone marrow MSCs and its correlation of the "MSCs" within the cartilage and periosteal bone is difficult to appreciate, as the context is very different. Can the author provide some assessment/comment on whether any of the marrow MSCs can contribute to the enchondromas as they have not eliminated such a possibility.

We commented on this in the revised manuscript.

The overall finding is very interesting. In the Discussion, the authors have provide what we already know concerning the role of HH and Wnt signaling in skeletogenesis; can the authors comment on what could be the differences in the action of HH when it is activated in "osteochondral progenitor cells" as oppose to differentiated osteoblasts and chondrocytes with respective to the relationship between HH and Wnt signaling leading the formation of tumors.

We commented on this in the revised manuscript (Discussion section).

Reviewer #3:The authors have addressed most of my experimental comments, which improves the paper. What remains and should be addressed for a high visibility paper is an extreme reliance on (1) review articles (for instance, 7 out of the first 16 references are review articles and it does not improve after) and (2) on citations of their own work or of their collaborators. This would be perfectly acceptable if they had done the original work but this is not the case. This needs to be addressed before going further.

We have removed 3 references authored by us and 13 other review-article references.